# Research on the Technological Progress of CZT Array Detectors

**DOI:** 10.3390/s24030725

**Published:** 2024-01-23

**Authors:** Zhangwen Li, Jinxing Cheng, Fang Liu, Qingbo Wang, Wei-Wei Wen, Guangwei Huang, Zeqian Wu

**Affiliations:** 1College of Nuclear Science and Engineering, North China Electric Power University, Beijing 102206, China; 120222212087@ncepu.edu.cn; 2Institute of Nuclear and New Energy Technology, Tsinghua University, Beijing 102200, China; wqb08@tsinghua.org.cn (Q.W.); wenwwep@126.com (W.-W.W.); huanggw@pku.edu.cn (G.H.); wuzeqian@novelmedical.cn (Z.W.)

**Keywords:** photon detection, CZT, array detector, technological progress

## Abstract

CdZnTe (CZT) is a new type of compound semiconductor that has emerged in recent years. Compared to other semiconductor materials, it possesses an ideal bandgap, high density, and high electron mobility, rendering it an excellent room-temperature composite semiconductor material for X-ray and γ-ray detectors. Due to the exceptional performance of CZT material, detectors manufactured using it exhibit high energy resolution, spatial resolution, and detection efficiency. They also have the advantage of operating at room temperature. CZT array detectors, furthermore, demonstrate outstanding spatial detection and three-dimensional imaging capabilities. Researchers worldwide have conducted extensive studies on this subject. This paper, building upon this foundation, provides a comprehensive analysis of CZT crystals and CZT array detectors and summarizes existing research to offer valuable insights for envisioning new detector methodologies.

## 1. Introduction

With the advancement of nuclear science and technology, nuclear detection technology is now widely used in people’s daily lives and scientific research. Examples include nuclear radiation monitoring, safer medical imaging, as well as a more effective non-invasive analysis for security, nuclear safety, or product inspection applications and radiation detection and imaging [1,2,3].

For public safety concerns and further exploration in nuclear science and technology, contemporary society is setting higher standards for nuclear radiation detection technology. Early gas detectors exhibited low detection efficiency for high-energy radiation, lacked high energy resolution, and had relatively larger volumes, making them only suitable for detecting medium to low-energy radiation. Scintillation detectors (such as NaI, CsI, MHPs, etc.), although inexpensive and having high detection efficiency for X-rays and γ rays, are limited in application due to their long response times, comparatively lower energy resolution compared to semiconductor detectors, and larger physical dimensions. Moreover, semiconductor materials, owing to their ability to operate at room temperature and their excellent resolution and faster response times, possess unique advantages in nuclear radiation detectors. These advantages have led to their extensive use in various fields such as national security, medical imaging, industrial non-destructive testing, astronomical observations, and cutting-edge physics [4,5,6].

In order to achieve a detector with high energy resolution, high detection efficiency for high energy radiation, and the capability to operate at room temperature, the compound semiconductor materials used in the fabrication of such detectors should possess specific physical properties. These properties include the following:Low ionization energy: To minimize the impact of statistical fluctuations, the semiconductor material should have a low ionization energy.Higher average atomic number: A higher average atomic number enhances the detector’s efficiency in detecting high energy radiation.Sufficiently wide bandgap: A wide bandgap allows the detector to function at room temperature with minimal leakage current.

Product of carrier mobility and lifetime (μτ): A high μτ product reduces the impact of carrier capture, improving the detector’s energy resolution [7]. In the context of detectors, the overall efficiency, energy resolution, and time response speed of the backend electronic components should be exceptionally high to effectively process data for image generation.

To meet these requirements, the technology of using Cadmium Zinc Telluride (CZT) semiconductor material to produce array detectors has been proposed and implemented. Detectors made of CZT semiconductor material are characterized by their high resistivity, wide bandgap, and ability to operate at room temperature. Such detectors have found extensive applications in a wide range of fields, including aerospace, astrophysics research, nuclear medicine, environmental monitoring, nuclear counterterrorism, and nuclear emergency response [8,9].

## 2. Development of CZT Material

### 2.1. The Basic Structure and Advantages of CZT Material

The CZT crystal can be regarded as an infinite solid solution crystal composed of CdTe and ZnTe crystals in a (1-x):x ratio, possessing a cubic zinc blende crystal structure, as illustrated in Figure 1. By varying the value of x, different proportions of CZT crystals can be obtained, leading to changes in their physical and chemical properties. This variation allows for the production of various types of CZT detectors, catering to the evolution of nuclear detection technology. Moreover, as the value of x changes, the bandgap width of the crystal varies between 1.49 and 2.26 eV.

In general, the fabrication of large-volume CdTe crystals is relatively easier compared to CZT crystals, with CdTe crystals achievable at a maximum diameter of 50 mm. Additionally, CdTe crystals exhibit good uniformity, devoid of grain boundaries, rendering CdTe relatively inexpensive. However, for CZT materials, the addition of Zn to the CdTe compound stabilizes the Cd-Te bond. Consequently, CZT crystals possess stronger inter-atomic covalent bonds compared to CdTe crystals, resulting in reduced susceptibility to defects [10].

By comparing CZT with other semiconductor materials, it is evident that during the crystal solid solution process, adjusting the Zn composition allows for changes in the bandgap width of CZT. This adjustment facilitates the preparation of CZT crystals with a larger bandgap width, granting CZT a wider bandgap compared to other semiconductor materials. As a result, CZT exhibits higher resistance and consequently lower leakage current in the detectors produced, leading to lower power consumption. Additionally, CZT’s higher average atomic number endows it with strong attenuation properties against γ radiation, high electron mobility, and excellent carrier transport characteristics, ensuring superior detection performance of CZT crystals for γ radiation [11].

In assessing the physical performance of semiconductor materials for radiation detection through statistical analysis, a comparative study was conducted to contrast and analyze the obtained results, as depicted in Table 1.

Through the comparison of the performance of various semiconductor materials in the table, it is evident that compared to other types of semiconductor materials, CZT crystals exhibit a well-balanced array of characteristics. With a larger atomic number and bandgap width, high resistivity, and rapid response rate, CZT demonstrates superior overall performance, making it highly suitable for nuclear radiation detection [3,15]. In comparison to other compound semiconductor materials, Cadmium Zinc Telluride (CZT) emerges as a highly competitive material for X-ray and γ-ray detection.

### 2.2. Preparation of CZT Materials

#### 2.2.1. The Evolution of the Way CZT Grows

The preparation methods for CZT can be broadly categorized into three main types: the melt method, the solvent method, and the vapor-phase transport method [16]. In order to produce CZT crystals with excellent properties and high uniformity, the commonly employed crystal growth methods include the Bridgman method (BV) and the Traveling Heater Method (THM) [17].

The Bridgman method is a practical technique characterized by a simple furnace structure and convenient operation. Essentially, it involves melting all CZT raw materials or polycrystalline materials and gradually solidifying the molten material from the head to the tail end by slowly moving the crucible or furnace. Moreover, based on variations in growth apparatus and growth conditions, it can be categorized into several methods including the High-Pressure Bridgman (HPB) method, Modified Vertical Bridgman (MVB) method, Horizontal Bridgman (HB) method, and the Vertical Gradient Freeze (VGF) method [18].

The HPB method is one of the earliest fabrication techniques. During the growth process, inert gas, typically argon, is used to compress the vapor pressure of the molten material into a small area, preventing element evaporation and safeguarding against atmospheric influences like water and oxygen. As early as 1992, Doty et al. [19] first utilized the HPB method and reported the viability of CZT as a γ-ray detector. Subsequent optimizations by Szeles et al. and the introduction of the High-Pressure Gradient Freeze (HPGF) technique moderately improved the monocrystalline rate of crystal production but still encountered crystal cracking issues. The HPGF method incorporated multi-segment temperature control and furnace temperature field improvements, successfully eliminating cracks. However, crystals grown by this method still exhibited polycrystalline and twinning phenomena, resulting in lower utilization [20,21,22,23].

The MVB method introduces several improved techniques based on the traditional Bridgman method, such as crucible rotation, additional Cd compensation, or the use of seed crystals for CZT crystal growth. In the MVB method, the introduction of seed crystals during CZT crystal growth resulted in Yinnel Tech, led by Li et al. [24], achieving 3-inch diameter CZT crystals, where a single crystal with a volume of approximately 100 cm^3^ accounted for roughly 80% of the entire crystal ingot volume. However, twinning and grain boundaries persisted within the crystal ingot. In 2007, Yadong et al. [25] investigated CZT crystals grown via the Modified Vertical Bridgman method. Testing CZT ingots doped with indium, the infrared transmittance remained relatively high and constant within the wave numbers of 2000 to 4000 cm^−1^. However, the transmittance rapidly decreased to zero between 2000 and 500 cm^−1^. These CZT crystals exhibited low dislocation density and high crystalline quality.

Wanqi et al. [26,27,28] initiated studies on CdTe/CZT crystals using the Bridgman method as early as 1993. In 2010, Wang Tao and Jie Wanqi, among others, utilized an improved Vertical Bridgman method to fabricate CZT crystals. With this method, crystals of over 60 mm in diameter achieved a utilization rate of over 70%. The crystal exhibited low tellurium precipitation and impurity density, measuring less than 1 × 10^3^ cm^−2^, while displaying a resistivity of 4 × 10^10^ Ω·cm. Detectors produced using these crystals exhibited excellent resolution for both low and high-energy radiation with a spectral resolution of 4.2% for ^137^Cs and 4.7% for ^241^Am γ spectra. In 2020, through the introduction of seed crystals and Cd compensation, along with crucible acceleration rotation and a series of technical optimizations, the team successfully achieved the Bridgman method’s full single-crystal growth of 2 inch CZT crystals for the first time internationally. This advancement has paved the way for industrial-scale implementation of the technology.

The HB method positions the ampoule horizontally during crystal growth, ensuring that the growth interface remains unaffected by the molten material’s weight. This growth technique offers a uniform matrix, enhances yield, and reduces production costs. However, a downside to this method, in comparison to the more traditional High-Pressure Bridgman (HPB) method, is that its volume resistivity is lower by an order of magnitude: approximately 10^9^ Ω·cm. This lower resistivity leads to higher leakage current, ultimately reducing the detector’s energy resolution at lower energies (<100 keV) [29].

The Traveling Heater Method (THM) involves a preparation technique where, from bottom to top, a seed crystal, a tellurium-rich melt, and CZT polycrystalline material are sequentially placed in the crucible. During crystal growth, the temperature field comprises a high-temperature zone in the melt region and low-temperature zones at both ends. Ideally, the melt region is thoroughly and uniformly mixed, ensuring the same composition and melting point at both the growth and dissolution interfaces. As the heater moves upward at a consistent rate, the lower growth interface’s temperature shifts toward the cooler direction, leading to supercooling and solute precipitation, while the upper dissolution interface’s temperature shifts toward the hotter direction, resulting in overheating and solute dissolution. With the slow movement of the heater, CZT solute continually dissolves from the dissolution interface into the tellurium-rich melt and is transported to the lower growth interface, where it precipitates, allowing for the continuous growth of CZT crystals [30].

In 2011, Wilson et al. [31] conducted research on CZT crystals grown using the THM for X-ray detection. They tested crystals sized at 80 × 80 mm^2^ with an array of 11 × 11 and a spacing of 250 μm. Their study validated that the combination of large area and low-density arrays can provide higher energy resolution, facilitating conditions for the spectroscopic imaging of high-energy X-rays. In 2013, Chen Xi et al. [32] utilized an electron probe and photoluminescence spectroscopy to test the radial composition and defect distribution of crystals prepared using the THM. They conducted a comparative analysis with CZT crystals grown using the Vertical Bridgman method. Their findings suggested that CZT crystals produced using the THM method exhibited greater uniformity, lower defect density, and better uniformity in defect distribution, especially in the case of large-volume CZT crystals. The uniformity of tellurium inclusions within the crystal significantly affects charge collection and charge transport [33].

In 2018, Chen et al. [34] from the University of Illinois at Urbana-Champaign fabricated large-volume single-crystal pixelated cadmium zinc telluride (CZT) detectors measuring 40 × 40 × 15 mm^3^ using the Traveling Heater Method (THM). The radiation sensor exhibited an energy resolution of 2.5% at 662 keV at room temperature. Following Depth of Interaction (DOI) correction, the pixelated 22 × 22 × 15 mm^3^ CZT detector achieved an energy resolution of <1% at 662 keV at room temperature. Simultaneously, the detector demonstrated better sensitivity, offering improved spectral performance for applications in nuclear and homeland security. However, challenges within the THM process involve slow growth rates and demanding temperature gradients. The high-temperature gradients induce vigorous liquid-phase convection, potentially leading to temperature fluctuations or uneven solute distribution, resulting in unstable growth interfaces and the potential for defect formation [35,36].

In 2021, Gao et al. [37] from the School of Optoelectronic Engineering of Chengdu University of Information Technology conducted a study on the preparation of large-area and high-crystallinity CZT crystals. A groundbreaking approach was employed, combining physical vapor transport and vacuum thermal evaporation (PVT-VTE) hybrid methods. This resulted in the fabrication of multi-crystalline CZT thin films with a dense pyramid surface structure, achieving a thickness of 590 μm. The pixelated array X-ray detector produced from these films exhibited an average dark resistivity of 3.33 × 10^11^ Ω·cm along the diagonal, while μ_e_τ_e_ is 0.72 × 10^−2^ cm^2^·V^−1^, which is close to the value of a single-crystal CZT device, showcasing high resistivity and sensitivity. This indicates that CZT thick films prepared by PVT-VTE will have great commercial application potential in the field of X-ray imaging and detection at room temperature in the future.

The summary evaluation of all preparation methods is presented in Table 2:

Different preparation methods for CZT crystal synthesis have their respective advantages and drawbacks, necessitating the selection of an appropriate manufacturing technique based on specific requirements and applications. Continuous technological improvements and optimizations further enhance the quality of CZT crystals, improving their responsiveness to radiation and advancing the application of CZT detectors in various fields.

#### 2.2.2. Doping and Performance Changes of CZT

Altering the component content within the crystal and introducing doping elements can modify the crystal’s performance. In 2003, Renchu et al. [38] researched the resistivity of CZT crystals grown under different excessive tellurium conditions. The study found that the most efficient detector was produced from CZT crystals containing a 1.5% excess of tellurium. This CdTe was adequate to fix the Fermi level in the middle of the bandgap without an excessive number of defects to capture charge carriers. Additionally, empirical research has shown that besides altering the ratio of Cd to Zn in CZT materials, minute elements introduced via other doping can also enhance crystal properties.

In 2012, Ruihua et al. [39] conducted a study on crystals doped with indium elements, achieving resistivity as high as 1 × 10^12^ Ω·cm. They utilized thermal stimulated current spectroscopy to measure the trap levels within CZT:In, determining that the high resistivity was due to changes in the crystal traps caused by indium doping, anchoring the Fermi level near the middle of the bandgap.

Researchers from Roy’s team at Brookhaven National Laboratory [40] investigated the impact of selenium addition to cadmium zinc telluride (CZT) matrix on radiation detector performance. Given that Zn segregation in CZT crystals can generate sub-grain networks, in 2015, Roy’s team initially utilized the MVB method to prepare selenium-doped crystals, which effectively guaranteed solid solution hardening in the CdTe matrix, avoiding sub-grain boundary networks in CdTe_x_Se_1−x_ crystals. They studied its properties and found that the etch-pit density in the crystals was one order of magnitude smaller than the commonly used CZT crystals, demonstrating improved charge collection efficiency in CTS compared to CZT.

In 2019, Roy et al. [41] synthesized a novel quaternary compound Cd_0.9_Zn_0.1_Te_0.98_Se_0.02_ using the THM and discovered that this crystal exhibited remarkable compositional uniformity with fewer extended defects such as second phases and sub-grain boundary networks. The virtual Frisch grid detector produced using this method displayed an energy resolution for 662 keV γ rays in the range of 0.87–1.5%. The study concluded that it possesses excellent material quality, extremely low defect density, and highly uniform composition, thereby presenting a promising prospect as an outstanding room-temperature detection material.

Roy et al. subsequently evaluated the CZTS detector for various γ ray energies [42]. At room temperature, the prepared CZTS detector crystal sized 4.5 × 4.5 × 10.8 mm^3^ exhibited extremely low leakage current: approximately 0.63 nA at a bias of 500 V. The Te resistivity of the detector was about 2.9 × 10^10^ Ω·cm, meeting the requirements for high resistivity detector-grade materials. Under irradiation by 662 keV γ rays, the detector achieved an exceptionally high resolution of about 1.07% at a bias of 3000 V. At 662 keV conditions, the shaping time was 2 μs, the energy resolution was within 1.2%, the peak-to-valley ratio was 28, and the peak Compton ratio was 5.6, demonstrating excellent detector performance. The energy resolution of this CZTS detector remained remarkably stable throughout its operation. For other γ ray energies such as 511 keV and 1.275 MeV irradiation, the energy resolution of the detector was about 1.34% and approximately 1%, respectively. Additionally, the detector successfully identified all γ ray lines from the 133 Ba source, including the 31 keV X-ray line, with an energy resolution of approximately 9%. For high-energy γ lines at 1.17 MeV and 1.33 MeV, the energy resolution of the CZTS detector was about 1.1% and 1.0%, respectively.

In 2017, Nan et al. [43] used an improved vertical Bridgman method to prepare Cd_0.9_Zn_0.1_Te crystals doped with aluminum under cadmium-rich conditions. The researchers observed the distribution of microscopic defects. The CZT:Al sample’s volume resistivity was measured to be 10^8^ Ω·cm. However, the resistivity of the material was not ideal, exhibiting relatively high leakage currents. The introduction of aluminum as a dopant led to an increase in crystal defects, consequently resulting in reduced detection performance.

Compared to other primary melt-growth methods, such as Bridgman-based techniques, the Traveling Heater Method (THM) is the most commonly utilized method for commercial CZT crystal growth. CZTS detectors fabricated using this method exhibit quality comparable to high-quality commercial CZT detectors. Additionally, doping with trace elements such as indium and selenium can enhance the crystal’s performance.

#### 2.2.3. Semiconductor Surface Treatment Technology

To enhance the performance of CZT detectors, the primary approaches involve establishing material selection standards, such as controlling lattice mismatch, and improving material surface defects. Additionally, refining the material preparation methods, including CZT annealing and MCT pre-growth preparation, can play a crucial role in these enhancements [44]. Since the development of CZT crystals for detector applications, researchers have extensively investigated the influence of surface treatments on the crystal’s performance. In 1999, Chen et al. from the University of Alabama [45] conducted a study on the passivation of CZT surfaces by low-energy atomic oxygen. They discovered that CZT samples exposed to low-energy oxygen atoms generated smoother and more uniform surfaces, thereby enhancing their performance and reducing leakage currents.

In 2006, Zhang et al. [46] treated mechanically polished CZT crystals with an H_2_O_2_ and N_2_H_4_ solution for different durations to passivate the surfaces. They observed that the surfaces became notably smoother and denser after passivation. The resistivity of the CZT increased, indicating that the surface passivation process significantly affected the performance of the CZT detector chips. Characteristic curve analysis and SEM surface morphology observations revealed that passivation with an H_2_O_2_ solution for 20 min showed better results. Additionally, passivation using an NH_4_F/H_2_O_2_ solution for 30 min notably reduced surface leakage currents, resulting in an increase in resistivity by 1–2 orders of magnitude to 10^9–10^ Ω·cm, demonstrating considerable improvement in reducing leakage current.

In 2010, Yubao et al. [47] conducted research on the surface-etching process during crystal preparation to reduce the surface leakage current of detectors. Testing revealed that oxidative etching at a certain power level not only removed the compositional segregation on the CZT surface between the electrodes but also generated a dense TeO oxide layer on the CZT surface, causing passivation of the electrode surface of the CZT pixel detector. Radiofrequency power was identified as a critical factor in the oxygen ion-etching process; using high power caused severe damage to the CZT surface, resulting in a substantial increase in surface leakage current. Conversely, when etching was performed at low power, chemical action dominated, leading to a sharp decrease in surface leakage current.

In 2014, Hossain et al. [48] improved the surface-etching process. They experimented with two types of reagents: a low-concentration bromine-based etchant mixture combined with a surface passivation reagent and a non-bromine-based etchant to treat the crystal surface. Ultimately, their developed new non-bromine-based etchant generated a non-conductive surface, reducing defects.

In the preparation process of CZT crystals, the impurities and micro-defects brought by the etching step are the main drawbacks of the CZT crystal, and they cannot be completely eliminated. These flaws affect the optoelectronic properties of CZT and the quality of the surface. Several techniques are used to suppress or eliminate the defects in CZT materials, and one of them is surface-coating treatment [49].

Yuan et al. [44] conducted a study on the fabrication processes of high-quality cadmium zinc telluride (CZT) crystal material and thin films of mercury cadmium telluride (MCT). They summarized methods for improving material quality and standard control. Through processes like heat treatment, they controlled the size of impurities and defects to be less than 20 μm with a density less than 20 cm^−2^. This ultimately increased the detector’s response rate to 2.6% and reduced the defective pixel rate to 0.03%. The improvement in crystal performance was achieved through high-quality CZT heat treatment techniques, and a method for preparing MCT-FPA with excellent uniformity imaging, contributing to an increased production and component capacity of existing products.

Zhang et al. [50] conducted research on the Chemical Mechanical Polishing (CMP) process of CZT crystals. From 2021 to 2022, the team aimed to obtain a high-precision CZT crystal surface by proposing a dual modification method for Stöber synthesis through improved technology. They carried out orthogonal experiments on the CMP process parameters and polishing liquid ratios of CZT crystals. The study explored the impact of factors such as abrasive quality fraction, H_2_O_2_ volume content, pH value, polishing pressure, polishing disc speed, and polishing liquid flow rate on the surface processing performance of CZT crystals. The results indicated that the abrasive quality fraction and polishing liquid pH value significantly influenced the polishing accuracy and processing efficiency of the CZT crystal surface. Gao [51] conducted comparative experiments based on the proposed new technology and new polishing liquid. They concluded that the super-precision surface processing performance of mesoporous structured abrasives was superior to that of solid abrasives. Also, the performance of the morphology and structure dual-modified RmSiO_2_ abrasives was better than that of single-modified abrasive grains.

Yu et al. [52] employed the Closed Space Sublimation (CSS) method to grow CZT films on (111)-oriented CZT wafers, non-oriented CZT wafers, and FTO substrates in 2022. The final results indicate that CZT films grown on (111)-oriented CZT chips exhibit low dislocation density and higher charge carrier transport performance. On (111)-oriented CZT chips, the electrical properties of the CZT film, including resistance and µ_e_τ_e_ values, were determined to be 9.4 × 10^9^ Ω and 7.3 × 10^−3^ cm^2^·V^−1^. It has good reactions to nuclear radiation signals and can detect the radiation from weak radiation sources. Therefore, CZT films grown on (111)-oriented CZT wafer are suitable for fabricating nuclear detection devices.

In summary, domestic and foreign improvements in semiconductor material surface treatment processes primarily focus on the enhancement of surface treatment technology and surface coating optimization. Improvements in heat treatment techniques and surface coating can effectively elevate the surface resistivity and electron drift mobility of detector crystals. Among these, the novel Closed Space Sublimation (CSS) method stands out as particularly effective in enhancing crystal performance, albeit with higher associated costs. However, post-optimization, it exhibits substantial commercial value.

### 2.3. Research on the Properties of CZT Materials

#### 2.3.1. Reaction of γ rays with CZT Material

The working principle of CZT as a detector material involves the interaction of high-energy radiation such as X-rays or γ-rays with the CZT crystal within a biased detector. When irradiated, the detector absorbs a portion or the entire energy of the incoming radiation. This absorption leads to the generation of electron–hole pairs within the crystal, which are proportional to the incident high-energy radiation. These pairs drift under the influence of an electric field toward different electrodes within the crystal. Eventually, they are collected on the electrodes [53]. Electric pulses are formed on the electrodes, and after passing through charge-sensitive preamplifier analog circuits, voltage pulses proportional to the energy of the incident radiation are output [54].

When high-energy radiation enters the CZT crystal and generates electron–hole pairs, the electric field within the crystal can be expressed by the formula, where *V_B_* represents the bias voltage applied to the crystal and *d* indicates the thickness of the crystal:(1)$$E=\frac{V_{B}}{d}$$

The charge drift velocity is shown below:(2)$$V=\mu E=\mu \frac{V_{B}}{d}$$

The induced current *i* can be obtained from Shockley–Ramo theorem using the following formula:(3)$$i=E_{Q}qv=\frac{1}{d}q\mu E=\frac{1}{d}q\mu E\frac{V_{b}}{d}=q\mu \frac{V_{b}}{d^{2}}$$

If the electron–hole pairs generated by the radiation in the CZT crystal reach opposite electrodes, the distance of charge drift equals the entire thickness of the crystal, which is denoted as *d*. Thus, when calculating the time for charge drift to reach the electrode, the charge collection time can be expressed by the following formula [55]:(4)$$t=\frac{d}{v}=\frac{d^{2}}{V_{B}\mu_{e}}$$

The induced charge *Q* of the detector is shown below:(5)$$Q=it=q\mu \frac{V_{B}}{d^{2}}\frac{d^{2}}{V_{B}\mu }=q$$

If the electron–hole pairs produced by the rays in the CZT crystal are at a distance of *x* from the anode, then the collection time of the electrons *t_e_*:(6)$$t_{e}=\frac{x}{v_{e}}=\frac{xd}{V_{B}\mu_{e}}$$

The time of collection of holes is *t_h_*:(7)$$t_{h}=\frac{d-x}{v_{h}}=\frac{(d-x)d}{V_{B}\mu_{h}}$$

The induced charges *Q_e_* and *Q_h_* generated by the two are, respectively:(8)$$Q_{e}=i_{e}t_{e}=q\mu_{e}\frac{V_{B}}{d^{2}}\frac{xd}{V_{B}\mu_{e}}=q\frac{x}{d}$$
(9)$$Q_{h}=i_{h}t_{h}=q\mu_{h}\frac{V_{B}}{d^{2}}\frac{(d-x)d}{V_{B}\mu_{e}}=q\frac{d-x}{d}$$

If the generation of electron–hole pairs occurs at the midpoint of the CZT crystal, then at the moment when the charges are fully collected: (10)$$Q_{e}=Q_{h}=\frac{q}{2}$$

Due to the migration efficiency of electrons being approximately three times that of holes *μ_e_* ≈ 3*μ_h_*, the charges collected on the electrode are not entirely complete when the electrons have fully arrived. This incomplete charge collection is due to the extended collection time of holes, resulting in tailing effects in the final output signal. The collection time of the detector is correlated with the detector’s thickness. Based on the specific thickness of the detector, an appropriate collection time can be calculated to facilitate the determination of specific subsequent circuit parameters. In 2009, Abbene et al. [56] conducted a study on the hard X-ray response of pixelated CZT detectors. Comparing detectors made in the same manner with identical materials but with CZT crystals of 1 mm and 2 mm thickness, the thicker detector exhibited better performance. This suggests that increasing the detector thickness improves both detection efficiency and charge collection, which is consistent with the small-pixel effect. To maximize the complete collection of generated charges within the lifetime of the hole, the charge collection time should be at least several times the electron collection time when the electron–hole pairs are at the midpoint of the detector. Determining the collection time provides a reference for establishing the pole-zero cancellation time and Gaussian shaping time, as these times must not be less than the charge collection time. Otherwise, it significantly affects the energy resolution of the detector system [57,58,59].

#### 2.3.2. Research on the Energy Resolution of CZT Materials

The energy resolution of a detector is a critically important performance indicator. As semiconductor materials form a crucial component of detectors, extensive research has been conducted on the energy resolution performance of CZT crystals.

In 2011, Wangerin et al. [60] studied three different geometries of CZT anodes and compared the energy resolution and absolute peak efficiency before and after data corrections. Following the correction of the test data, both the energy resolution and efficiency showed improvement. It was observed that detectors with a pixel pitch of 1.3 mm, exhibiting the best small-pixel effect, presented superior energy resolution, with an optimal resolution of 5.1 keV [61] Beginning in 2008, Bolotnikov et al. from the Brookhaven National Laboratory [62] conducted research on extended defects in CZT crystals for γ-ray detectors. The CZT material contains a high concentration of extended defects, especially tellurium inclusions, dislocation networks, grain boundaries, and sub-grain boundaries, affecting the device’s energy resolution and efficiency. In 2014, Huichao et al. [63] from Shanghai Jiao Tong University investigated the noise performance of CZT crystals. They studied the detector’s geometric shape, detector voltage, and time constants. The incomplete charge collection within the detector crystal is the main factor affecting detector performance. The primary reason is that the product of the mobility and lifetime of holes in the material is one to two orders of magnitude smaller than that of electrons. Bolotnikov et al. from BNL [64] conducted research on charge loss under normal circumstances, as shown in the following formula:(11)$$Q=Q_{0}exp(-\frac{t_{drighft}}{\tau_{bulk}})$$

*Q* and *Q*_0_ are the charged and initial charges collected by the incident particle or photon, *t_drigft_* is the drift time of the electron cloud from the interaction point to the anode, and *τ_bulk_* is the lifetime of the electrons in the CZT body. When a defect occurs inside the crystal, the charge is lost as shown below:(12)$$Q=Q_{i}\{1-\eta_{i}\left[1-\exp\left(-\frac{t_{i}}{\tau_{i}}\right)\right]\}$$

*t_i_* and *τ_i_* represent the electron drift distance and lifetime; they are averaged over the *i* geometric region. $\eta_{i}$ is a portion of the charge of the electron cloud passing through the encompassed geometric area. The integrated charge left behind in the electron cloud after encountering it is shown as:(13)$$\;Q=Q_{i}\{1-\eta_{i}\left[1-\exp\left(-\frac{D_{i}}{E_{i}\mu_{i}\tau_{i}}\right)\right]\}$$

*E_i_* is the local electric field strength, *μ_i_τ_i_* represents the attenuation distance, and *i* denotes the average distance.

It has been observed through testing that the decrease in collected charge signal amplitudes due to tellurium inclusions is proportional to the electron cloud’s drift distance. 

Based on this, in 2015, the domestic team led by Chuang [65] characterized the prepared CZT pixel detectors. They analyzed the impact of tellurium inclusions and other defects within the crystal on the electronic properties, such as detector leakage current and spectral response characteristics. The comparison revealed that an enrichment of tellurium inclusions increased detector leakage current, reduced charge collection efficiency, and deteriorated the detector’s energy resolution. In 2019, Lei et al. [66], in an effort to enhance CZT’s discrimination capability, introduced an improved method using ^10^B as a conversion layer for thermal neutron detection. The reaction of neutrons with ^10^B primarily generates energies of 2.792 MeV and 2.31 MeV: n + ^10^B → α + ^7^Li + 2.792 MeV (6.1%)(14)
n + ^10^B → α + ^7^Li* + 2.31 MeV (93.9%)(15)

The 93.9% neutron capture is accompanied by the γ emission of 0.482 MeV energy, which makes the energy resolution of the designed detector reach 5.25%, and it has good neutron detection performance, but the disadvantage is that its n-γ shielding ability needs to be improved.

To achieve a higher energy resolution in CZT crystals, it is necessary to reduce the crystal’s leakage current. This can be accomplished through surface treatments, optimized electrode designs, minimizing grain boundaries and defects, controlling impurity levels within the crystal, selecting appropriate sizes, and using high-quality crystals. These optimization methods are effective at enhancing the energy resolution of CZT crystals, making them more suitable for high-precision detector applications [67]. 

The thickness of the CZT itself also has an effect on the energy resolution of the detection system. In 2014, Min et al. [68] conducted experiments and simulations on CZT crystals of different thicknesses using various radiation sources. They found a high level of consistency between theoretical and experimental values of energy spectra and peak efficiencies for CZT detectors of different thicknesses under different γ photon energy conditions. They noted that thicker CZT detectors achieved higher energy resolution and peak efficiency at higher energies, whereas thinner detectors exhibited relatively better characteristics at lower energies. Thin CZT crystals at 662 keV, using ^137^Cs as a radiation source, showed low photon attenuation characteristics. The conclusion drawn was that for detecting high-energy radiation, CZT crystals must have sufficient thickness to provide better accumulation of high-energy photons and improved detection efficiency. In 2021, Hui et al. [69] from Northwestern Polytechnical University studied the impact of crystal thickness on the spectrum of CZT detectors. They found that as the CZT thickness increased, the peak height of the final image increased, the full width at half maximum (FWHM) decreased, the peak address decreased, and the total charge collection efficiency reduced.

In summary, to improve the energy resolution of detectors by enhancing CZT crystals, key methods include optimizing the geometric shape of the array to achieve the best small-pixel effect. Additionally, improving crystal performance and enhancing energy resolution can be achieved by changing the semiconductor crystal thickness and optimizing manufacturing processes.

#### 2.3.3. Research on the Energy Bands of CZT Crystals

For the semiconductor crystal, when the energy of incident photons is greater than the bandgap width, electrons in the valence band are excited into the conduction band. An interaction between electrons and the crystal lattice causes electron transitions into the conduction band. The vacancies left by the departing electrons, known as holes, then transition back to the valence band. When the exciting photons are removed, the excess electrons and holes cannot maintain a thermal equilibrium state within the crystal. Consequently, they recombine. Electrons transition from the conduction band back to the valence band. As the scattering of photons must adhere to the conservation of energy and momentum, the scattered energy equals the difference in energy between the conduction and valence bands [70]. 

It has been found that Cd_1−x_Zn_x_Te just like CdTe is also a direct bandgap crystal. Because ZnTe (E*g* = 2.26 eV) has a larger bandgap than CdTe (E*g* = 1.46 eV), so the inclusion of Zn elements will make the bandgap of Cd_1−x_Zn_x_Te larger than that of CdTe, and the bandgap will change with the increase in zinc content. There are some differences in the results obtained by different researchers.

The relationship between zinc content and bandgap energy levels in the summarized studies is presented in Table 3:

For different temperatures T, the bandgap width of CZT also varies. At *x* = 0.5, the relationship between the bandgap width and temperature can be expressed by Formula [79]:E*g* (eV) = 1.72 − 3.5 × 10^−4^T − 1.55 × 10^−7^T^2^(16)

When the energy of incident photons exceeds the bandgap width, electrons in the valence band are excited to the conduction band. The interaction between electrons and the crystal lattice induces electron transitions to the conduction band. Vacancies left behind by the departing electrons, known as holes, subsequently transition back to the valence band [80].

In 2015, Rongrong et al. [81] investigated the impact of sub-bandgap light on CZT crystal. They discovered that under external sub-bandgap illumination, the crystal’s defect density decreased, leading to a more uniform electric field distribution and a reduction in the concentration of active traps. Additionally, measurements of the ^241^Am γ ray spectrum response by the detector confirmed that simultaneous illumination with sub-bandgap light significantly enhanced the energy resolution and charge collection efficiency of CZT detectors.

In 2022, Xiang from the China Institute of Atomic Energy [82] conducted an optimization of the energy band structure of crystals, utilizing a CZT crystal with dimensions of 5 mm × 5 mm × 1 mm provided by Shaanxi Detai Technology. Their study focused on the optimization of the γ sensitivity performance of CZT detectors, specifically addressing the issue of overshoot in the calibration current curves for CZT detector γ sensitivity. To address the trapping effect of deep-level traps on charge carriers within the crystal, the study employed sub-bandgap light to adjust the occupation of deep-level traps by charge carriers, thereby eliminating the overshoot in the calibration current curves of the detector. This approach increased the charge carrier collection efficiency, leading to the optimization of CZT detector performance. As a result, the particle sensitivity of the detector to γ rays with an energy of approximately 1.25 MeV reached 1.18 × 10^−16^ C·cm^2^.

Optimizing the detector sensitivity by enhancing the energy band structure of CZT crystals has been proven to hold practical value. From the aforementioned studies, it is evident that increasing the bandgap and reducing crystal defect density can effectively enhance the energy resolution and detection efficiency of detectors; further research in this area holds potential for additional exploration and investigation.

## 3. Development of CZT Array Detectors

### 3.1. Different Structured CZT Detector

#### 3.1.1. CZT Detector Structure

The development history of semiconductor nuclear radiation detectors spans back to the 1960s and is continuing to progress significantly. As early as 1949, researchers discovered that Ge diodes produced a pulse output when exposed to α particles. Subsequently, a variety of radiation detection materials and devices were introduced. Presently, commonly used semiconductor radiation detectors mainly include the PN junction detector, silicon lithium-drifted detector, high-purity germanium (HP-Ge) detector, and compound semiconductor detectors. In the realm of compound semiconductors, significant research focuses on materials like GaAs, CdTe, CdZnTe, and SiC, which are CZT materials that hold significant positions [83].

The configurations of photodetectors vary and can be broadly classified into two main categories based on their photocurrent mechanisms: photodischarge devices and photovoltaic devices. Photodischarge devices feature a structure with an anode and cathode forming an ohmic contact MSM (metal–semiconductor–metal) configuration. Generally, the active region in this structure is the surface layer that receives the incident radiation. Its advantage lies in its relatively better photoresponse in short wavelengths. However, due to the presence of ohmic contacts, this configuration may exhibit higher dark currents and lower device sensitivity as well as slower response speeds [84].

Photovoltaic devices consist of various structures such as the Schottky diode structure, MIS structure, PN junction structure, PIN junction structure, and avalanche photodiodes (APDs) [85]. In comparison to photodischarge detectors, the active region of photovoltaic detectors penetrates deeper into the semiconductor. This allows for thickening the active region through structural design or increased reverse bias, thereby enhancing the device’s quantum efficiency. Due to the presence of the Schottky diodes or PN junctions, these devices typically exhibit lower dark currents and faster response speeds [86]. Based on their operational principles, photovoltaic devices can be further divided into two categories: depletion type and avalanche type. Depletion-type devices involve the process of photon-to-electric conversion occurring in the depletion layer with the photocurrent primarily generated by the drift of photo-generated charge carriers. Common examples include Schottky diodes, PN junctions, and PIN structures. On the other hand, avalanche-type devices, as implied by the name, operate in the avalanche region, where photocurrent gain is primarily generated through the avalanche effect, as seen in APDs. While these devices offer significant internal current gain, they demand strict requirements in terms of materials and device properties [87]. Illustrations of various photodetector configurations are depicted in Figure 2.

In materials like CZT, the electron mobility is approximately 1100 cm^2^/V·s with a migration lifetime of around 2 μs, whereas the hole mobility is about 100 cm^2^/V·s with a migration lifetime of approximately 0.05 μs. The significant disparity in the mobility abilities of electrons and holes implies that holes are captured extensively. As per the Hecht equation (Formula (17)), the induced charge collected by the electrodes is dependent on the distance traveled by the corresponding charge carriers within the crystal. The substantial capture of holes results in incomplete hole signals collected by the cathode. In contrast, due to their longer migration lifetime, electrons can be fully collected by the anode. This asymmetry in collection leads to an asymmetric signal peak, shifting toward the lower energy region. Consequently, this effect, known as the hole tailing effect, reduces the energy resolution [89].
(17) Q=N0e[λed1−e−xcλc+λed1−e−xhλh]

*N*_0_—number of electron–hole pairs;d—detector thickness;*λ*—the drift length of the electron with the hole;*x*—the length of the drift that the electrons must do with the holes.

To address the issue of low energy resolution caused by the hole tailing effect, research primarily employs two main categories of methods: correction of pulse signals through electronic methods and adjustments made in the electrode structure to achieve unipolarity [90].

The adjustments made in the electrode structure primarily rely on unipolar charge induction. According to the Shockley–Ramo theorem, the induced charge on the electrode is directly proportional to the change in the weighted potential as the charge q moves from the starting point to the ending point [91,92].

The adjustment in the electrode structure aims to eliminate the electrical signals generated by hole drift, thereby focusing on collecting the signals generated only by electron drift to enhance energy resolution. Presently, the primary electrode structures for CZT detectors include planar, Frisch grid structure, coplanar grid structure, pixelated structure, hemispherical detector, quasi-hemispherical detector, and others as illustrated in Figure 3. 

The planar structure is the simplest configuration, as depicted in Figure 3a. In a planar CZT detector, metal is deposited on both sides of the CZT crystal, forming a metal–semiconductor–metal structure. When high-energy radiation penetrates the CZT crystal and exceeds the material’s bandgap, it excites electrons from the valence band to the conduction band, thereby creating electron–hole pairs. Under the influence of the electric field from the electrodes, electrons and holes are collected to induce charge. However, this method faces issues as the electron mobility in CZT is significantly higher than that of holes. Consequently, holes are more susceptible to capture by impurities and defects within the crystal, resulting in a non-uniform collection of electrons and holes at the electrodes [94].

The Frisch grid structure, illustrated in Figure 3b, involves creating a Frisch ring electrode near the anode position on the side of the CZT detector and grounding it. Under a specific external bias, the electric field in the device primarily exists between the grid and the anode. Given that radiation photons enter from the cathode and stop within the CZT near the cathode, it is apparent that only electrons passing through the annular surface of the Frisch grid can go through the electric field region and induce charge. This design excludes any induced signals generated by charge movement between the cathode and the Frisch grid, allowing the collected signals from the device to be considered as generated solely by electron drift. This effectively eliminates the hole tailing effect [95].

The coplanar grid structure, as shown in Figure 3c, involves creating two sets of interdigitated comb-like strip electrodes on the anode, applying different voltages to these two grid electrodes. The grid with the higher potential is referred to as the collecting electrode, while the one with the lower potential is known as the non-collecting electrode. This structure generates a stronger electric field near the anode, leading to a rapid change in the weighting potential. Conversely, the weighting potential change near the cathode tends to be more gradual, thereby reducing the electric signals produced by hole drift near the cathode. As the distance between the coplanar grid electrodes decreases, the charge collection becomes more uniform. However, a smaller electrode distance may result in higher surface leakage currents in the device [96].

The hemispherical detector is one of the earliest unipolar semiconductor radiation detectors with an ideal structure depicted in Figure 3d. The quasi-hemispherical and hemispherical structures are similar in principle. Due to the difficulty in manufacturing true hemispherical crystals, a simplified cubic shape is often used to represent the hemispherical structure, as illustrated in Figure 3e. All four lateral surfaces and the bottom surface act as cathode surfaces, while the anode contact is situated in the center of the anode surface. This structural setup results in a strong electric field near the anode with a steep rise in the weighting potential. Meanwhile, the electric field and weighting potential near the cathode are relatively flat, effectively reducing the impact of hole drift on the electric signal generation process [97].

Array-type detectors, illustrated in Figure 3f, consist of a grid of equally sized squares forming the anode surface. Each square represents a small pixelated anode. During operation, the pixelated anodes are connected to a high voltage, while the cathode plane is grounded. As electrons move away from the anode, all pixel arrays produce weak signals. When electrons move closer to the anode, the signals of the other pixel detectors gradually decrease, while the signal from the corresponding pixel detector position continually increases until reaching its maximum value. Similarly, the weighting potential changes slowly near the cathode and dramatically increases near the anode [98]. It is important to note that the smaller the pixel size, the higher the spatial resolution achieved, but this also increases unipolarity. However, this comes at the cost of reduced signal-to-noise ratio and decreased charge collection efficiency, which is known as the small pixel effect. Similar to the configuration of a coplanar grid, reducing the pixel size is also accompanied by a significant increase in surface leakage current [99].

Among all unipolar structures, the pixelated configuration performs the best and has the most mature processes. The performance of the detector unit is directly related to the size of the pixel; smaller pixels exhibit better unit performance, which is a phenomenon referred to as the “small pixel effect” [100]. Pixelated CZT detectors are commonly used in two-dimensional imaging and, when combined with depth sensing techniques, allow for three-dimensional imaging. This integration enhances energy resolution, offering a wide array of potential applications [101].

#### 3.1.2. Operating Principle of the CZT Array Detector

The operating principle of the CZT array detector involves segmented electrode design on one side of the CZT crystal, creating a pixel array electrode structure on the crystal’s anode surface, while the cathode comprises a solid plane electrode. When radiation enters from the cathode surface, a negative high voltage is applied to the cathode surface. The ionizing radiation interacts with the crystal, generating numerous electron–hole pairs. Under the influence of the electric field formed by the applied bias, the electrons and holes migrate toward their respective poles. The induced charges are collected by the pixel electrodes. They are then converted into voltage pulse signals proportional to the amplitude and energy of the incident photons by a charge-sensitive preamplifier. Subsequently, shaping amplifiers further amplify these signals to enhance the signal-to-noise ratio. Finally, the signals can be utilized to generate corresponding spectral distributions or, through signal acquisition and processing, to create images based on the pulse amplitude distribution statistics [102].

The radiation enters from the cathode surface while applying a negative high voltage on the cathode surface. The ionizing radiation interacts with the crystal, creating numerous electron–hole pairs. Under the influence of the externally applied bias, the electric field is established. Electrons and holes migrate toward their respective poles, generating induced charges collected by the pixel electrodes. These charges are converted by the charge-sensitive preamplifier into voltage pulse signals proportional to the amplitude and energy of the incident photons. Subsequently, these signals are further shaped and amplified by a shaping amplifier to achieve higher signal-to-noise ratios. Finally, the signals can be used to derive the corresponding spectral distributions through statistical analysis of the pulse amplitude distribution or processed to generate images after signal acquisition and processing [103]. The schematic diagram depicting its operation is illustrated in Figure 4.

The electron–hole pairs move toward different electrodes under the external bias of the detector and are eventually collected. These signals can be used to form the energy spectrum of incident photons via a multi-channel analyzer. In the study of array detectors, the assessment criteria for the detector’s performance include noise, energy resolution, response speed, and other aspects. These criteria are interdependent, mutually constraining one another. Therefore, it is important to individually analyze various influencing factors.

### 3.2. Research on the Performance Optimization of CZT Array Detectors

#### 3.2.1. Optimization of Energy Resolution

The energy resolution refers to a detector’s actual ability to distinguish nuclear information and is generally defined as the smallest relative difference between the energies of two adjacent values for a given energy. Due to statistical fluctuations in the detection process, even for a single nuclear radiation energy, the relationship between the collected count rate and energy is not a straight line but a distribution curve. Therefore, the Full Width at Half Maximum (FWHM) of this curve, indicating the width of the curve at its half maximum, is often used to represent the characteristics of resolution [104].

When CZT detectors operate at room temperature, the leakage current is one of the primary parameters determining the detector’s performance. As unipolar detectors only collect signals generated by electron carriers, excluding the impact of holes, the signal induced by electrons in the detector’s electrodes is proportional to the total charge generated by photon deposition. The effect of holes depends on the detector’s structure and the generation position of electron–hole pairs. It has been demonstrated that uncollected holes can affect the detector’s performance, reducing the energy resolution, particularly evident in planar detectors. Smaller anode arrays can decrease the influence of holes, enhancing collection efficiency and energy resolution while also achieving relatively higher spatial resolution with a simple structure [105].

In 2005, the research team led by Bolotnikov [106] investigated the primary factors influencing the energy resolution of CZT detectors. The study explored three key factors that limit the performance and ultimate energy resolution of CZT devices:Impact of leakage current: In CZT devices with Au and Pt contacts, the overall leakage current is restricted by the characteristics of the Schottky barrier at the metal–semiconductor interface.Influence of charge sharing among pixels: Inter-pixel electric conduction affects the distribution of electric field lines, leading to charge loss between adjacent anode contacts in multi-electrode devices.Effects of charge loss: Charge loss usually accompanies charge sharing. Some electrons in the electron cloud between pixels fall into the gap and remain uncollected by the pixel electrodes, resulting in charge loss.

In 2012, the team [107] conducted research on the energy resolution of CZT detectors, improving the application-specific integrated circuits (ASIC) and data acquisition systems originally used for 3D pixel detectors. They introduced a universal cathode readout system to correct charge signals and eliminate incomplete charge collection events. The study evaluated a 2 × 2 size, 6 × 6 × 15 m^3^ virtual flash grid detector array and found that in applications requiring lower position and energy resolution, large-pixel, low-density array formats might replace more advanced but costlier 3D pixel array detectors. Furthermore, the large-pixel, low-density array offered improved energy resolution, higher stopping power, and position resolution. It was demonstrated that by employing a cathode signal readout system, incomplete charge collection events due to crystal defects could be filtered out. Thus, it became feasible to utilize crystals with some defect content at a lower cost for manufacturing such arrays [108].

In 2007, Wilson et al. [109] measured the signal shapes produced by alpha and X-ray radiation in a 2 mm thick CZT detector. They compared the signals generated by a single large substrate detector and a 300 μm pixelated detector. Eventually, they used TCAD simulation software to directly compare experimental data, allowing for the visualization of carrier motion within the CZT detector. This visualization helped determine that the primary cause of charge sharing events is the initial size and subsequent diffusion of the carrier cloud.

In 2011, the team led by Jiang et al. [110] conducted research on the noise performance of cadmium zinc telluride (CZT) array detectors. They employed low-noise fast preamplifier modules, analyzed the internal performance of the crystal, and investigated the impact of the preamplifier on the noise of the detection system. They established a 2 × 2 cadmium zinc telluride pixel array detection system. Experimental results showed that the output signal noise of the detection system was minimal with no pulse pile-up. The electronic noise of the readout circuit was significantly suppressed, resulting in reduced low-energy tailing in the spectra. Improvements were noted in the noise caused by incomplete charge collection and leakage current noise. The team also conducted a noise analysis of the entire system and the preamplifier, proposing relevant improvement methods that effectively suppressed the electronic noise of the readout circuit.

In 2022, Wei et al. [111] designed critical circuits in both single-channel and sixteen-channel readout circuits to mitigate the impact of leakage current on the detector’s energy resolution. The single-channel readout circuit comprises CSA, SHAPER, DIS, and PDH modules. Additionally, to suit the specific application scenario of SPECT, they added leakage current compensation circuits and baseline holding circuits based on the traditional single-channel design. They further designed a loop bias module, enhancing the electronic design of the detector. As a result, they achieved a maximum compensation of 50 nA for leakage current, a peaking time of 150 ns, a channel gain of 50 mV/fC, less than 1% integral non-linearity, and a maximum injection frequency of 500 kHz.

At the charge sharing optimization realm, in the last century, Bolotnikov et al. [112] discussed the charge loss between pixel detector contacts and studied several different gap contact arrays’ charge loss on a CZT detector. They identified the maximum contact gap at which the charge loss between surface pixels could be neglected. They discovered that the minimum signal loss occurred with a contact size of 450 μm and a distance of 50 μm between contact edges. As the array gap increased, there was a rapid increase in signal loss, and there was the appearance of pixel channel dead zones in the array gap [113]. In 2007, the research group also designed a hexagonal grid CZT crystal virtual flash grid detector to evaluate the performance of the CZT detector’s basic design from the perspective of the residual effect of uncollected holes in the CZT detector. In practice, it is not feasible to entirely shield the charge caused by uncollected holes in the entire active volume of the device. However, in the CZT detector’s design, it is possible to correct the output signal changes in the drift region while rejecting interaction events from the induction region.

In 2007, Iniewski et al. [114] proposed an analytical model to provide an effective framework for studying the influence of detector geometry, bias conditions, and material properties on detector performance. The model accurately predicted the number of charge-sharing events as a function of photon energy and detector pixel size. Simulations were performed on a material with a Cd_0.9_Zn_0.1_Te composition. Comparative results revealed that a higher number of sharing events occurred when the gap between 0.46 mm-sized pixels was larger than 0.1 mm and increased with the radius “*r*” of the electron cloud. Additionally, it was determined that corner pixels within the array exhibited fewer sharing events compared to edge pixels, while edge pixels exhibited fewer sharing events compared to center pixels. The energy resolution (FWHM) was poorest for corner pixels, followed by edge pixels, with the center pixels demonstrating the best energy resolution. In 2009, Yin et al. [115] characterized high-resolution CZT detectors for sub-millimeter PET imaging. Their research discovered that the ratio of events between central single-pixel events and central double-pixel events decreased when the radiation energy increased from 59.5 to 122 keV. As the pixel array size became very small (350 μm), the impact of charge sharing on detector energy resolution might be more critical than the small pixel effect. The team measured the distribution of charge-sharing events and combined them with collimated beam size and their contribution to charge sharing. Under detector conditions with a 600 μm spacing and 5 mm thickness and a 350 μm spacing, the optimal gap for charge sharing in the measurement of a 122 keV collimated beam was approximately 125 μm.

In 2011, Veale et al. [100] discussed the charge-sharing effects in small-pixel CZT detectors. They employed ASIC software developed by the Rutherford Appleton Laboratory to compare the amount of charge sharing under various anode geometric shapes. Their study concluded that in comparison to array detectors with the same pixel size but different spacings, the shape of the final spectrum is influenced by increased charge sharing in the anodes. Consequently, arrays with larger spacings exhibited a decrease in the amplitude of the main peak relative to the lower energy peaks.

In 2007, Bolotnikov et al. [116] discussed the performance of 20 × 20 × 10 mm^3^ and 10 × 10 × 10 mm^3^ single-pixelated detectors as well as a 4 × 4 × 12 mm^3^ virtual flash grid device. They detailed various physical properties of the materials and proposed the concept of “small gaps” as a solution. Their findings revealed that reducing the gaps between pixels effectively reduces charge loss. When the gap size is reduced to below 100 μm, it achieves an effect similar to that of the guard ring.

As the crystal array’s reduced spacing could result in the occurrence of charge-sharing events, in 2011, Kim et al. from the University of Michigan proposed an enhancement approach for pixelated cadmium zinc telluride (CZT) detectors by designing a guiding ring. This involved the addition of extra electrodes between the gaps, as illustrated in Figure 5.

The design of the guiding ring typically encircles each pixel, to some extent improving the edges. To ensure that the electron cloud (charge carriers) between the pixels is collected by the pixel electrodes under the influence of the electric field, the guiding ring’s width needs to be minimized while applying adequate negative high voltage. However, this introduces surface leakage currents between the pixel electrodes and the guiding ring, leading to increased electronic noise in the pixel anode and consequently a reduction in energy resolution to some extent. This method presents challenges in fabrication. Imperfect electrode processing can result in low surface impedance, causing excessive surface leakage currents and potentially leading to certain pixel channels becoming non-functional. In the absence of sufficient bias, significant charge loss phenomena might occur.

In 2023, Mele et al. [105] designed a high-energy-resolution CZT linear array pixel detector. They utilized a passive filtering circuit, incorporated internal (500 µm wide) and external guard rings around the pixels to minimize charge-sharing events, and employed quasi-ohmic gold contacts to ensure low leakage current at room temperature. The experimental results demonstrated excellent spectral performance with a full-width at half-maximum (FWHM) of 258 eV (4.34%) at 5.9 keV, 576 eV (0.97%) at 59.5 keV, and 1.17 keV FWHM (0.96%) at 122 keV. The detector exhibited a higher-than-expected response, indicating significant limitations on the original resolution imposed by the transport characteristics of the detector’s material and by the geometrical dimensions of the pixel.

In conclusion, to enhance the overall energy resolution of the array detector, improvements are required not only in the crystal performance but also in the channel-level performance of the detector. Enhancing the contact methods between various components, such as by refining the readout circuitry and implementing guiding rings, reduces the loss during charge transfer, thus elevating the final energy resolution.

#### 3.2.2. Research on the Spatial Resolution of Array Detectors

High-energy radiation imaging detection has been a highly researched topic in various fields such as high-energy nuclear physics, astrophysics, and nuclear security detection both domestically and internationally. A 3D position-sensitive CZT room-temperature γ-ray spectrometer typically comprises an array detector crystal with independent pixelated anodes and dedicated channels for signal processing. Each channel includes an integrated circuit for readout, incorporating a preamplifier, a shaping amplifier, and a sample-and-hold device. The depth of γ-ray penetration is determined by the ratio of the cathode to anode signals from each pixel [118]. For obtaining high-quality images, the primary technical requirements for array detectors include high detection efficiency and sensitivity for high-energy or γ radiation. Additionally, a large signal dynamic range, indicated by the ratio of the maximum output signal (open-circuit signal) to the system’s noise, is necessary. Good isolation between pixels is crucial to eliminate signal interference among pixels, and appropriate pixel size is essential [119].

In 2009, a collaborative effort between NASA and the Lawrence Livermore National Laboratory (LLNL) [120,121] led to the establishment of a large-area 128 × 128 pixel 32 cm × 32 cm high-energy CZT pixel array imaging detection system. This system covered the energy range from 10 to 600 keV. It achieved an energy resolution of 5.37% for ^241^Am and a spatial resolution of 2.5 mm.

In 2012, Zhang et al. [122] characterized detectors using a new application-specific integrated circuit (ASIC) developed by the Brookhaven National Laboratory’s Instrumentation Division. Their findings revealed that the energy resolution of three-dimensional position-sensitive CZT detectors does not necessarily decrease with an increase in detector volume/thickness. The excellent energy resolution of the detector indicates that large-area CZT detectors can approach the theoretical limits of energy resolution.

In 2012, Yin et al. [123,124,125] conducted a study on three significant factors influencing the three-dimensional spatial resolution of 350 μm pitch CZT array detectors. These factors included charge sharing, intrinsic spatial resolution measurements, and Depth of Interaction (DOI) analysis. They found that with an increase in γ-ray energy, the number of charge-sharing events in double-pixel arrangements notably increased. This indicates a larger charge cloud size and a higher probability of Compton scattering resulting from higher energy γ-rays. Additionally, they observed a linear relationship between the γ-ray electron drift time and the cathode/anode ratio.

In 2018, Ukaegbu et al. [126] from the School of Engineering at the University of Glasgow in the UK developed a decay model for cadmium zinc telluride (CZT) detectors. This model aimed to estimate the depth of remotely buried radioactive waste. Through a comparison with an organic liquid scintillator detection system, the established model was capable of estimating the depth of a 329 kBq ^137^Cs radioactive source buried within a 12-centimeter-thick layer, yielding an average count rate of 100 counts per second. Experimental validation using a 9 kBq ^60^Co radioactive source affirmed that the model could be applicable for any γ-radiation source. Furthermore, it demonstrated the ability to estimate the depth of buried sources with extremely low activity.

In 2022, a team led by Pan Chao from the China Academy of Launch Vehicle Technology [127] proposed a method for implementing a multi-beam array detection three-dimensional imaging lidar system. Their study involved research into this system and its performance as well as the analysis of errors through computer simulation. This work aimed to provide a theoretical foundation for the design and parameter optimization of multi-beam array detection three-dimensional imaging lidar systems.

In 2000, Zhong et al. from the Department of Nuclear Engineering and Radiological Sciences at the University of Michigan [128,129] introduced a general technique to address carrier capture issues, including hole and electron capture. This method involved determining the Depth of Interaction (DOI) information for each event, followed by the correction of different positions’ charge collection efficiencies, resulting in uniform charge collection efficiency throughout the entire detector. This approach is known as Depth Sensing and Correction (DOI correction). When implemented in pixel detectors, it can determine three-dimensional information about the interaction position. Pixel detectors are typically designed for imaging. To acquire depth information, Wen et al. [130] proposed a signal modeling method used to calibrate the relationship between C/A values and the depth of interaction. The process involved assuming equal gains for cathode and anode electronics, depositing energy at different depths, calculating the corresponding cathode and anode signal amplitudes, and finally plotting the relationship between C/A values and the depth of interaction.

In 2006, Liptac et al. [131] employed fast digitization and software signal processing techniques using the HXR diagnostic method to image the energy of a CZT detector system containing 32 arrays within the 20–200 keV range. By comparing spectra between different channels, they obtained spatial information about fast electron clusters.

In 2010, Vernon et al. [132] introduced an improved application-specific integrated circuit (ASIC) designed for Three-Dimensional Position Sensitive Detectors (3D PSD). By altering the anode channels to process two polarity events simultaneously and store amplitudes in the corresponding positive and negative peak detectors, they addressed the issue of additional counts above the photopeak for energies higher than the light peak. By increasing the number of anode channels to 128 while maintaining symmetrical layouts, the new detector required two ASICs to read out a 256-pixel sensor, measuring peak amplitudes and relative timings for 128 anodes, one anode, and the cathode. The shaped analog signals from each channel could be multiplexed to an auxiliary output for monitoring purposes. The multiplexing and readout logic was optimized to reduce dead time and achieve higher count rates.

In 2021, Lee et al. [133] utilized Geant4 simulations in combination with an enhanced median Wiener filtering technique and edge detection methods to improve the quality of the fused γ-ray and X-ray images obtained from CZT detectors. They demonstrated that the method combining MMWF and edge detection algorithms showed superior filtering performance in γ-ray and X-ray fused images produced by photon-counting CZT detectors compared to traditional methods.

In three-dimensional CZT detectors, the amplitude of induced signals depends on the Depth of Interaction (DOI). Therefore, the calibration of detectors using the Depth of Interaction correction technique plays a crucial role in improving the energy resolution of γ-ray detectors. Li [134] from Nanjing University of Aeronautics and Astronautics conducted research on existing DOI correction methods and proposed an improved energy-correction algorithm. The experimental study discussed DOI correction factors for CZT detectors at various energy levels. By utilizing a segmented energy-correction method, the research significantly improved the energy resolution and peak height of multiple energy peaks in the energy spectrum, achieving good correction results in multi-energy γ-ray detection. Furthermore, the research extended the DOI correction method for use in Compton imaging γ detectors, resulting in a noticeable enhancement in the image intensity for Compton imaging.

To enhance the spatial resolution of CZT array detectors, the primary research focus lies in refining the signal readout circuitry and optimizing algorithms, particularly emphasizing improvements in energy calibration algorithms. Such enhancements hold significant importance for improving the spatial resolution of the detector and the overall quality of imaging.

#### 3.2.3. Optimization of Detection Efficiency

Zhang et al. from the China Institute of Atomic Energy [135] developed and simulated a large-area array neutron-γ detection system based on plastic scintillator materials for specific nuclear material detection applications. They utilized Geant4 for initial structural optimization by simulating key detector components. The specific workflow for structural optimization is illustrated in Figure 6: 

The simulation involved comparing the detection response threshold and the n/γ discrimination ratio before and after applying different thicknesses of lead and tungsten for shielding. The comparison revealed that within a certain range, increasing the thickness of the shielding material gradually increased the n/γ discrimination ratio. Due to lead’s lower ability to attenuate neutrons compared to tungsten, different thicknesses of lead provided a higher n/γ discrimination ratio. The simulation optimization results indicated that for a plastic scintillator with a 25 μm thick Gd_2_O_3_ layer, and when the detection response threshold was set at 3, with a 3 mm thick lead plate between each layer of detectors, the system achieved approximately 23% neutron detection efficiency and an approximate 8/2 n/γ discrimination ratio. This ratio was roughly twice as high as that without any shielding material [136]. This implies that improving the surface structure of the array detectors and adding shielding layers can enhance detection efficiency and the n/γ discrimination ratio.

In 2009, Zhang et al. [137] conducted research focusing on low-noise and stable performance thick-film circuits. A CZT detector with a thickness of 1 mm exhibited a photoelectric absorption efficiency of 97.4% for 60 keV gamma rays, but this efficiency decreased to 58.3% for 100 keV. As the CZT detector thickness increased to 4 mm, the photoelectric absorption efficiency for 100 keV improved to 97.07%. However, for higher-energy 662 keV gamma rays, there was a noticeable decrease in detection efficiency. Even with a detector thickness of 15 mm, the photoelectric absorption efficiency only reached 23.6%. CZT detectors with an appropriate thickness can efficiently detect photon energies in the range of 200 to 300 keV.

In 2017, Fan et al. [66] used ^10^B as a conversion film and conducted a simulation analysis of CZT crystals using the MCNPX software. They found that the total detection efficiency and alpha (α) detection efficiency of the crystal gradually increased within the range of coating thickness from 0 to 1.6 μm, reaching a peak at a thickness of 1.6 μm. At this point, the total detection efficiency was 4.55%, while the α particle detection efficiency was 3.63%. This demonstrates that CZT requires a coating thickness of only 1.6 μm to achieve maximum detection efficiency, offering conditions for manufacturing compact portable neutron detectors.

In 2018, Fayuan et al. [138] studied the influence of crystal thickness on the detection efficiency of CZT detectors. Due to the challenges in growth technology, it is difficult to produce larger single crystals greater than 1 cm in size with high-quality cadmium zinc telluride (CZT), resulting in increased costs. The efficiency of crystal detectors in detecting X-rays and γ-rays depends on the crystal’s thickness. For instance, a 1 mm thick CZT detector has a photoelectric absorption efficiency of 97.4% for 60 keV γ-rays, which reduces to 58.3% for 100 keV γ-rays. When the detector thickness increases to 4 mm, the photoelectric absorption efficiency for 100 keV γ-rays improves to 97.07%. However, there is a notable decrease in detection efficiency for higher energy 662 keV γ rays. Therefore, CZT detectors are generally limited to detecting X-rays and γ-rays in the medium to low-energy range (10~600 keV), making it challenging to efficiently detect high-energy photons.

The study investigated the impact of cadmium zinc telluride (CZT) thickness and coating on crystal performance. By analyzing the response to γ-ray spectra, it was observed that in the higher energy range, increasing the thickness of the detector crystal to a certain extent or layering could result in improved detection efficiency and peak efficiency compared to a single-layer detector. The Compton continuum could also be improved, offering performance close to that of an entire CZT detector with equal thickness. However, the collection efficiency of photo-generated charge carriers decreased. The research demonstrated that the preparation of multilayer CZT detectors or the optimization of detector crystals to enhance detection efficiency is feasible but requires a comprehensive consideration of various impacts.

### 3.3. Array Detector Electronics

#### 3.3.1. CZT Crystal Contact Electrode

As the initial link in the signal transmission of a detector, the interface characteristics between the semiconductor crystal and the electrode play a crucial role in the detector’s performance. The mutual diffusion between the metal electrode material and CZT, along with the defects at the interface, are important areas of study.

Regarding electrode contact, as early as 2000, Bolotnikov et al. [139,140] conducted research on the influence of the geometric shape of the contact position on charge collection. They employed an orthogonal thin strip contact design between pixel contacts, applying a negative bias on the grid relative to the pixel potential to induce charge drift toward the contact point. This reduced the impact of charge loss between pixels, without significantly increasing leakage current, thereby enhancing the overall energy resolution of the detector [141].

In 2004, Wen et al. at Shanghai University [142] prepared CZT crystals with electrode layers composed of Au, Al, and In on both sides. After comparison, it was determined that the chemically inert Au-sputtered electrode is more likely to form quasi-ohmic contact on p-type high-resistivity CZT compared to other related electrodes, resulting in better crystal resistivity.

In 2010, Liang et al. [143] conducted a study on the electrode contacts of CZT detectors. They examined the contacts of Au-CZT and Au/Cr-CZT through stress simulations and multi-channel spectral analysis. The results demonstrated that using Au/Cr composite electrode materials can enhance the adhesion strength and thermal stability of the electrodes.

In 2016, Roy et al. [144] investigated the surface treatment of Cd_0.9_Zn_0.1_Te detectors by depositing AZO (Al:ZnO) thin films on the surface using atomic layer deposition (ALD) technology. The study affirmed that AZO serves as a promising alternative contact layer for CdTe based room-temperature-operating nuclear radiation detectors. Zinc oxide offers advantages such as higher hardness, better chemical stability, and closely matched thermal expansion coefficients with CdTe/CdZnTe, among other qualities. Detectors with AZO contact points demonstrated improved sensitivity and higher spectral performance.

In 2019, Ling et al. [145] conducted a study on the impact of Au/Cd composite electrodes on the conductivity of CZT crystals. The results indicated that the direct deposition of Au electrodes on the crystal surface caused lattice damage to the CZT, forming an amorphous layer, which was unfavorable for ohmic contact formation. The study found that the addition of a Zinc (Zn) intermediate layer at the Au/Zn-CZT interface was beneficial in eliminating the amorphous layer and promoting the formation of ohmic contact. Furthermore, the deposition of a Cd layer on the surface of CZT chips resulted in the formation of a CdTe buffer layer, aiding in the elimination of the negative impact of Te-rich layers on the Au/CZT contact, thereby enhancing the electrode performance.

In summary, the interface properties of the contact electrodes in semiconductor crystals play a crucial role in the performance of detectors. The selection of different electrode materials and their geometrical shapes significantly impacts the charge collection efficiency and energy resolution. Some studies have shown that using composite electrode materials, such as Au/Cr, can achieve better quasi-ohmic contact, thereby enhancing the electrode’s adhesion strength and thermal stability. Additionally, methods involving the introduction of intermediate layers or the replacement of contact layers with thin films of zinc oxide (AZO) can improve the performance of CdTe-based nuclear radiation detectors, enhancing detector sensitivity and spectral capabilities. These investigations provide valuable insights into electrode design and performance optimization in CZT semiconductor materials, potentially driving further advancements in semiconductor detector technology.

#### 3.3.2. Readout Circuitry

The CZT pixelated array detector system primarily comprises the X-ray or γ-ray sources, the CZT detector, a high-voltage power supply capable of providing stable voltage, the readout circuitry for analyzing and processing the weak signals outputted by the detector, analog-to-digital converters for data acquisition, and PC-based image processing software [146]. The front-end analog signal readout electronics system consists of the charge-sensitive preamplifier, which amplifies the weak charge signals from the detector, the shaping amplifier that filters and shapes the output pulse signal into quasi-Gaussian waveforms, analog-to-digital converters for data acquisition, and PC-based image processing software [147].

In a CZT pixel array detector, a corresponding number of charge-sensitive preamplifier circuits are required for each pixel to perform the preamplification processing of the weak electrical signals collected by each individual pixel. Subsequently, these signals are shaped into quasi-Gaussian waveforms by filtering and shaping amplification circuits. The information obtained from these procedures is then processed by a multi-channel analyzer to derive the energy spectrum of the incident photons [148].

When spectral detection is required, there are primarily two approaches for the design of the detector’s readout circuitry. The first involves the use of a design based on mature large-scale integrated circuits. This approach is associated with higher costs, longer development times, and more demanding technological requirements. The second approach employs a design based on discrete components for the readout circuit. This method is characterized by lower costs and greater design flexibility, allowing continual refinement of the design during the circuit’s design and production process, which can result in relatively good technical specifications. However, circuits produced by this method have lower integration levels and larger physical volumes [149].

The electrical signal output from the detector upon exposure to X-ray and γ-ray radiation is preamplified by a preamplifier and transformed into an exponentially decaying signal with a long time constant. This pulse signal, after passing through a pole-zero cancellation circuit, undergoes a downswing elimination, narrowing its width. The processed signal then passes through multiple stages of filter circuits for shaping, converting the pulse signal into a quasi-Gaussian waveform [150].

2007, Gevin [151] designed a 32-channel front-end ASIC called IDEF-X Eclairs to optimize the low capacitance and reduce the dark current of the readout circuit. The developed chip had a readout capacitance of 2–5 pF, dark current ranging from 1 pA to several nA, excellent noise performance, and high immunity to interference, demonstrating potential for space applications.

In 2014, Gao et al. [152] used a 0.35 μm CMOS process to design an ASIC circuit and conducted tests. The detector exhibited an energy resolution of 5.9% (FWHM) for a radioactive source of ^241^Am. They also evaluated the Total Ionizing Dose (TID) radiation effects using ^60^Co as the radiation source, demonstrating the ASIC’s ability to withstand 200 krad (Si) TID irradiation.

In 2017, Espagnet et al. [93] aimed to enhance the performance of CZT detectors by investigating the layout, front-end electronics, and characteristics of the dual-channel anode geometry. They determined the compensating coefficients required to reduce electron trapping in CZT, thereby reducing the number of channels needed for using the crystal in high-sensitivity counting applications. The study also investigated the impact of cathode voltage on detector performance. Ultimately, the optimal charge collection and minimal leakage current were achieved with a cathode voltage (UHV) of −2047 V, resulting in a virtual coplanar detector with 34% intrinsic efficiency and 8% energy resolution at a bias of 662 keV.

In 2008, Hao et al. [153,154] developed a compact 16-channel nuclear electronics array detection system optimized for low-energy γ rays, utilizing small-sized CZT semiconductor detectors. They integrated the system’s working principles with the test results. At room temperature, for 59.5 keV γ-rays from ^241^Am, each detection channel exhibited an overall noise of about 8.9 keV (FWHM) with a count rate of approximately 9 × 104/s. In the context of large-volume spatial detection, in 2010, the team used MCNP and other software to simulate strategies for enhancing Compton scattering detection efficiency and overall detection quality based on array detection principles.

MCNP stands for Monte Carlo N-Particle transport code, which is a versatile code used for simulating the transport of neutrons, electrons, and photons, allowing the coupled transport of these particles. It can handle complex 3D geometries and is known for its general purpose capabilities and flexibility. The team conducted research on the number of scattered photons at different angles between the incident radiation and the detection surface. They used the distribution of scattered photons to determine the optimal placement of the detector and optimized the collimator structure. The study revealed that effective scattered detection occurs when the incident angle of the radiation on the object being examined falls between 20° and 40° and the scattering angle of the scattered radiation ranges from 40° to 150° [155].

In 2012, Liu et al. [156] conducted a study on the readout circuitry for CZT array detectors used in nuclear radiation detection. They designed a charge-sensitive preamplifier tailored to the output signal characteristics of the CZT detector, converting its charge-type output signal to a voltage signal. They further designed a two-stage inverting operational amplifier to linearly amplify the weak voltage signal. Subsequently, they developed a single-capacitor correlated double sampling circuit that effectively eliminated noise from the signal. Through simulations, their designed readout circuitry was capable of handling radiation in the energy range of 100 to 500 keV, demonstrating low power consumption while effectively extracting, linearly amplifying, denoising, and converting the output charge signal from the CZT detector. Their work laid the foundation for the single-chip integration of CZT nuclear radiation pixel array detectors.

In 2019, Wei et al. [157] focused on the development of CZT detector systems and designed and implemented a 12-bit 1 MS/s resistive–capacitive (RC) hybrid structure successive approximation register analog-to-digital converter (SAR-ADC) chip. The chip demonstrated a power consumption of 10 mW and an area of 1.274 mm^2^. It features a simple and easily implementable circuit structure, making it suitable for CZT detector systems in portable gamma-ray spectrometers.

Based on the foregoing, people have never ceased their research and innovation in detector readout circuitry. Novel readout circuits applicable in various scenarios are consistently being developed. There is an overall trend toward precision and portability optimization, laying the groundwork for the lightweight design of future specialized nuclear detection devices.

#### 3.3.3. Charge-Sensitive Preamplifier

Preamplifiers used in nuclear detectors and measurement systems are mainly categorized into three major types: charge sensitive, voltage sensitive, and current sensitive. Among these, the charge-sensitive preamplifier is extensively utilized in high-resolution energy spectrum measurement systems due to its stable output gain, low noise, and excellent performance. Key performance parameters used to evaluate preamplifiers include noise, sensitivity, resolution, stability, rise time, and dynamic range. Of these, noise stands as a pivotal indicator and is illustrated in the schematic diagram in Figure 7 [158].

In the design setting of a fixed channel, gain is a design criterion, and the amplification factors of the Charge-Sensitive Amplifier (CSA) and SHAPER are derived from Equation (18) to establish the ideal value for the feedback capacitance, *C_f_*:(18)$$\frac{{di}_{n}^{2}}{d_{f}}=2{qi}_{d}+\frac{4kT}{R_{d}}+\frac{4kT}{R_{f}}$$

Formula (19) represents the expression of the equivalent current noise $i_{n}^{2}$ at the input of the Charge-Sensitive Amplifier, *i_d_* stands for the detector leakage current, *R_d_* is the parasitic resistance of the detector, and *R_f_* and *C_f_* represent the feedback resistance and feedback capacitance of the CSA, respectively. *q* is the elementary charge amount, having a value of 1.6 × 10^−19^ C. *k* is the Boltzmann constant, and *T* is the absolute temperature [159].

The equivalent current noise at the output end of the CSA is given as shown below:(19)$$\frac{{di}_{CASO}^{2}}{d_{f}}=\frac{{di}_{n}^{2}}{d_{f}}\frac{1}{\omega^{2}c_{f}^{2}}=\frac{{di}_{n}^{2}}{d_{f}}\frac{1}{{\left(2\pi fC_{f}\right)}^{2}}$$

According to the derived formula, achieving a specific value for the equivalent capacitance at the input of the CSA leads to better noise performance [160].

#### 3.3.4. Research in Signal Processing

In order to improve the γ energy spectrum of detectors, Xianyun [148] from Tsinghua University employed numerical simulation methods in 2005 to systematically study the γ-energy spectrum response of CZT detectors. They investigated the impact of various detector geometric parameters, incident radiation energy, different incident angles, and different filter layer thicknesses on the detector’s γ-energy spectrum response. Based on a familiarization with traditional γ-spectrum analysis methods, they fully applied new spectrum analysis concepts and algorithms to enhance the spectrum resolution. By using a qualitative approach with neural network methods, they quantitatively analyzed the actual measured γ spectrum of CZT detectors. The results showed that OLAM network spectral resolution exhibited high accuracy, fast computation, and ease of use, making it suitable for portable γ-spectrometers. Furthermore, they further explored a method for distinguishing special nuclear materials in γ spectra by suggesting that the technique of reconstructing γ spectra using direct demodulation could effectively differentiate radioactive isotopes such as ^239^Pu and ^238^Pu.

In the realm of signal shaping and processing, researchers like Krummenacher and colleagues from MIT [161] in 1991 achieved high-temperature plasma detection in nuclear reactors using a linear array CZT detector. The detection system utilized a pinhole approach, achieving a spatial resolution of 14–17 mm.

In 2010, Xin et al. [162] conducted research on pulse-shaping circuits for CZT detectors. They incorporated S-K low-pass filters into signal processing for CZT detectors, utilizing high-speed operational amplifiers (LMH6702). The resulting shaping circuit displayed excellent characteristics, enabling the acquisition of good waveforms with fewer stages.

In 2017, Zhao et al. [163] proposed a rapid imaging algorithm for MIMO radar based on a two-dimensional CZT sparse array. This algorithm utilized CZT transformation to replace interpolation operations, integrating compressed-sensing techniques for sparse array imaging, effectively reducing computational load and enhancing image focusing.

Wei from Northwestern Polytechnical University [157] delved into array detector imaging. From 2015 to 2017, this team focused on the design of Successive Approximation Register–Analog-to-Digital Converters (SAR-ADC) for array detectors. They successfully developed three SAR-ADC chips tailored for CZT detector systems employed in portable γ-spectrometers, PET biomedical imaging with CZT detectors, and space-based X/γ-ray detection. These chips were characterized by high precision, radiation resistance, and multi-channel integration.

In summary, the research on the electronics of CZT array detectors mainly aims to obtain better γ spectral information and to obtain more accurate results by optimizing the algorithm and improving the readout circuit and detector system.

## 4. Application of CZT Array Detector in Nuclear Detection and Imaging

### 4.1. Application of CZT Detector in Nuclear Detection

In the detection of special nuclear materials, non-contact detection methods are used to detect radioactive isotopes using X-rays, and γ-passive non-destructive analysis characterized by X-ray spectroscopy is the preferred method [164]. High-purity germanium (HPGe) detectors have traditionally been the first choice for such applications. However, they require cooling through liquid nitrogen or thermoelectric methods, which can be inconvenient for portable equipment. As a result, the demand is increasing for γ-ray measurement devices that can operate at room temperature [165]. Three-dimensional array CZT, as a high-energy-resolution, room-temperature-operable, deployable γ-ray imaging spectrometer, is capable of detecting and characterizing Special Nuclear Materials (SNMs) [166]. The CZT detector exhibits unique capabilities for the detection of SNM due to its ability to locate fast neutrons and γ-rays, its sensitivity to thermal neutrons, and its <1% γ-ray energy resolution at room temperature. This positions it with a growing scope of applications in this field [167,168].

In 1998, Zhong et al. at the University of Michigan [169] collaborated with the Johns Hopkins University Applied Physics Laboratory to upgrade two three-dimensional position-sensitive CZT spectrometers. They assembled a prototype Compton scattering γ-ray imaging device using the two upgraded CZT detectors. The individual performance of the two γ-ray spectrometers was independently tested. The angle resolution and detection sensitivity of the imaging system were measured using point and line sources of ^137^Cs radiation. The measurements matched the results from Monte Carlo simulations, confirming the potential of room-temperature three-dimensional position-sensitive CZT detectors in nuclear detection. In 2001, Mortreau et al. [170] investigated the characterization of cadmium zinc detectors’ spectra for spent fuel analysis using room-temperature semiconductor detectors. The respective detectors have been employed in the field of nuclear security for characterizing spent fuels (SFATs), enabling determination of the purity of enriched uranium and verification of the radiation status of nuclear fuels.

In 2015, Bolotnikov et al. [171,172,173] developed a position-sensitive Virtual Frisch-grid (VFG) CZT detector array based on a 2 × 2 array. They combined this with a robust detector module designed by Brookhaven National Laboratory’s readout ASIC to assemble a large-area, high-energy-resolution γ-ray detection instrument. It was demonstrated that the detector can be divided into sub-arrays, where their cathode connections form a single electrode, enabling signal correction for electron charge loss within the VFG detector and rejecting incomplete charge collection events. The design included position-sensitive edge contacts to correct material defects within CZT detector-grade crystals, significantly improving the acceptance rate of useful CZT crystals and reducing detector costs.

In 2020, David Goodman and collaborators from the Idaho National Laboratory [174] conducted research on the detection of ^239^Pu/^240^Pu isotopes using a digital 3D position-sensitive CZT detector. They used the digital CZT system to detect the isotopic composition and effective percentage of ^240^Pu in nuclear materials. Through statistical measurements from reactor-grade to weapons-grade nuclear material samples, the plutonium and uranium content exhibited reasonable consistency with uncertainties in the statistical measurements less than 3*σ*. The designed digital system is applicable for quantitative analysis in γ-ray spectroscopy.

In 2021, Xuesong et al. [175] analyzed the characteristics of γ-ray energy deposition in three key processes that can form Compton edges. They primarily used cadmium zinc telluride (CZT) detectors as the main detectors and bismuth germanate scintillators as anti-coincidence detectors. They designed a new structure for a portable anti-Compton γ detector suitable for measuring strong radiation sources. The Geant4 program was used for simulating and calculating the measurement spectra of 662 keV and 1525 keV γ-rays in two detection systems. The results showed that the overall external dimensions of both systems were controllable. The 4π-structured detection system had Compton suppression factors of 63 for 662 keV and 29 for 1525 keV γ rays, while the quasi-4π structured detection system had suppression factors of 51 for 662 keV and 26 for 1525 keV γ-rays. The simulation results were in line with the design specifications, validating the feasibility of the system.

The above findings demonstrate the vast potential of CZT detectors, particularly in the field of nuclear detection, especially for detecting special nuclear materials.

### 4.2. Research on the Application of Compton Imaging and Positioning

#### 4.2.1. Principle of Image Formation

Now, the application of CZT array detectors is currently focused on Compton imaging and localization. The concept of a Compton camera was initially proposed by Shenefeld in 1973 and subsequently tested in various applications, particularly in high-energy astrophysics and environmental radiation measurements, such as detecting radioactive elements in soil and open environments. The principle is that when γ-rays undergo Compton scattering with extranuclear electrons, angle information regarding the source of the radiation can be inferred from the scattering signal, allowing for the estimation of the location of the radiation source [176].

The scattering angle range of Compton scattering is 0°–180°, and the probability of Compton scattering lines produced in different scattering angle directions is referred to as the differential cross-section, which is expressed by the following equation [177]:(20)$$\frac{d_{\sigma }}{d_{\Omega }}=\frac{r_{e}^{2}}{{2[1+\alpha (1-\cos\theta )]}^{2}}\{1-{cos}^{2}\theta +\frac{\alpha^{2}{(1-\cos\theta )}^{2}}{1+\alpha \left(1-\cos\theta \right)}\}$$

*R_e_* represents the classical electron radius, *α* is a constant, *d_σ_*(*θ*) is the probability of one of the photons being scattered into a differential solid angle *d*_Ω_ at an angle *θ* from the incident direction. *D*_Ω_ is the differential solid angle in the direction of the scattering angle *θ*; *r_e_* and *α* are, respectively, like [178]:(21)$$r_{e}=\frac{e^{2}}{m_{0}c^{2}}=2.818\times 10^{-13}\;cm$$
(22)$$\alpha =\frac{E_{0}}{m_{0}c^{2}}$$

The formula mentioned is the Klein–Nishina formula [179], where *e* represents the charge of an electron, *m*_0_ is the rest mass of an electron, and *c* denotes the speed of light. According to calculations, the higher the energy of incident photons, the lower the probability of Compton scattering. The ratio between events of high-angle Compton scattering to low-angle Compton scattering also decreases as the incident photon energy increases. This implies that higher-energy photons are less likely to undergo Compton scattering and are even less likely to undergo large angle Compton scattering.

When a γ-ray from a radiation source undergoes Compton scattering once within the detector, it deposits energy *E*_1_ at a specific position x_1_ in the detector, and the detector records the scattering position as x_1_. The energy of the scattered photon produced by Compton scattering is detected a second time, depositing energy at position x_2_. The total energy deposited by the γ-ray in the detector is *E*_2_. The positions x_1_ and x_2_ where the γ-ray deposits energy twice, as well as the magnitudes *E*_1_ and *E*_2_ of the two energy deposits, can be obtained using position-sensitive cadmium zinc telluride detectors. *E*_0_ represents the energy of the γ-ray and can be directly obtained by adding *E*_1_ and *E*_2_. With the information obtained from the position-sensitive detector regarding *E*_1_ and *E*_2_, combined with the Compton scattering formula, the scattering angle *θ* can be calculated as follows [180]:(23)$$E_{c}=\frac{E_{0}}{1+\frac{E_{0}}{m_{0}c^{2}(1-\cos\theta )}}$$

As the Compton scattering formula cannot deduce azimuthal information, the position of the radiation source must lie at a specific point on the surface of a conical section, where the vertex of the cone is at x_1_, the axis is x_1_x_2_, and the half-angle is *θ*. After several occurrences of events that satisfy these conditions, many cones can be obtained. Theoretically, the position of the radiation source lies at the intersection of these cones. For instance, after detecting three valid events, three cones can be obtained. If the rays originate from an ideal point source, the three cones will inevitably converge at a unique focal point. This allows the determination of the radiation source’s position, as illustrated in Figure 8: 

If we assume that the incident γ-ray deposits an energy *E*_1_ in the Compton scatter and the remaining energy *E*_2_ in the absorber,

The scattering angle *θ* is calculated as follows:(24)$$\cos\theta =1-[\frac{m_{e}c^{2}E_{1}}{E_{2}\left(E_{1}+E_{2}\right)}]$$

However, in practical scenarios, both position and energy measurements are not entirely precise. To more accurately depict real-world conditions, cadmium zinc telluride detectors are typically divided into numerous grid units. When a γ-ray deposits energy within one of these units, the position of the energy deposition can be approximately represented by the center position of the cadmium zinc telluride detector grid unit.

#### 4.2.2. Developments in Imaging Technology

The data collected by the detector ultimately comprise countless discrete events, containing numerous invalid events, which cannot directly represent the position of the radiation source. To express information more intuitively, it is necessary to employ image reconstruction methods to process the data, transforming it into accurate visual representations. The primary methods used include analytical reconstruction algorithms, iterative algorithms, and maximum likelihood estimation/expectation maximization methods (MLEMs) [180].

In 2004, Zhang et al. [122,181] optimized the resolution and detection efficiency of the developed CZT3D detectors. They also conducted detailed research on the correction of time walk, concluding that to minimize time walk, the trigger threshold for the cathode should be set as low as possible, while the anode threshold should be set higher.

In 2007, De Geronimo et al. and Zhong et al. [182,183] extended the maximum likelihood expectation maximization (MLEM) method to array systems in spatial and spatial–energy domains. By utilizing the interaction event system response function of a new computational model, they demonstrated the spatial domain maximum likelihood expectation maximization for interaction events. The MLEM method was employed using standard iterative MLEM equations to find the most probable source distribution in either spatial or combined spatial–energy domains, resulting in image formation. Multiple sets of restored images revealed that as the number of interaction events increased, the three-dimensional images obtained from detection became more realistic.

After data analysis, it was discovered that when performing MLEM deconvolution on two or more interacting events, it is possible to distinguish the energy peaks of different atomic numbers of substances irradiated by γ-rays in the γ-ray spectrum, achieving material discrimination.

In 2010, Miao et al. [184,185,186,187,188] established a γ-ray pinhole imaging detection system using a 39 mm × 39 mm CZT pixel array detector. They analyzed the response characteristics of the CZT detector to the 662 keV ^137^Cs source, obtained the system’s overall modulation transfer function through ^137^Cs point source response imaging, and acquired and analyzed degraded images of the ^137^Cs source at the center and edge positions of the CZT detector. The detection images were restored using bilinear interpolation and the Lucy Richardson algorithm. Through a discussion and analysis of the system’s modulation transfer function and additional noise characteristics, it was determined that the pixel aperture transfer function and the diffusion effect of charge carrier signals in the detector were the primary factors limiting the spatial resolution of the CZT high-energy imaging detection system. Therefore, it is possible to enhance the pixel aperture transfer function by improving the electrode fabrication process and further reducing the pixel electrode size. Additionally, by introducing protective gate electrodes between pixels, the goal of suppressing charge carrier signal diffusion can be achieved. In 2011, an imaging evaluation model based on the impact of trapped charge carrier induction was established for CZT pixel array detectors. The detector’s modulation transfer function and pre-sampling modulation transfer function were derived. The study explored the influence of different key physical parameters on the imaging performance of CZT detectors under strong electric fields, with promising simulation results.

In 2017, Jianqiang et al. [189] conducted research on aspects such as pixel electrodes and readout methods for the 3D CZT detection system. They carried out experimental studies aimed at acquiring information on interaction depths, intending to achieve three-dimensional position sensitivity, spectral reconstruction, and imaging technology. In response to the demands of the detection system, the team developed the data acquisition system from both hardware and software perspectives.

The hardware design entailed creating a data acquisition board that fulfilled the following functions:Power Supply Output: provision of DC power configuration to the ASIC module as required; supply of the necessary high-voltage power to the cadmium zinc telluride detector.Data Communication: response to control and configuration commands from the PC; transmission of the current system status and acquired data to the PC.ASIC Configuration: configuration of 650 internal registers within the ASIC module based on configuration commands from the PC.Trigger Threshold Setting: adjustment of the trigger voltage of the ASIC module according to configuration commands from the PC.Conditioning and Digitization of Energy and Time Signals: energy information output by the ASIC module in the form of differential current signals necessitates analog conditioning to convert it into voltage signals before analog-to-digital conversion. Time information from the ASIC module is in the form of voltage signals, requiring initial driving before analog-to-digital conversion.Readout Timing Control: control of the ASIC module for data readout according to the corresponding timing circuits.Self-Testing Functionality: testing the status of the Ethernet connection.

The schematic diagram illustrating the principle design of the data acquisition board is depicted in Figure 9: 

In the software domain, the team developed upper-level PC data acquisition software that corresponded to the hardware. This software was designed to facilitate system monitoring, data acquisition, and analysis for the 3D CZT detection system. The data acquisition software was required to possess the following functionalities:Configuration and monitoring of the current operational status of the data acquisition board;Real-time visualization of collected data for quick diagnosis of the detector’s operational status;Capability to save the collected data.

Researchers developed 3D CZT detection system data acquisition software using LabVIEW8.6 software. This software featured three distinct panel interfaces, each corresponding to system configuration, real-time display, and file storage. The software underwent testing and adjustments to ensure its functionality and effectiveness.

In 2018, Kim et al. from the Korea Institute of Radiological and Medical Sciences [190] developed a Compton backscatter imaging detector to enhance the sensitivity of portable non-mechanical collimation detectors and achieve three-dimensional imaging. Using the ML-EM algorithm for full-width at half-maximum (FWHM) analysis of the reconstructed images, the detector demonstrated 3D imaging capabilities and the ability to penetrate several tens of centimeters deep into the object under examination.

In 2014, Yufei et al. [191,192] conducted research on Compton backscattering imaging (CBST) using an improved iterative reconstruction algorithm. Addressing the sensitivity of CBST image reconstruction to noise and error in measured values, they proposed a novel CBST reconstruction algorithm. This method involved the use of the augmented Lagrangian multiplier to decompose the optimization problem into two sub-problems with analytical solutions. By iteratively solving these sub-problems, the augmented Lagrangian function was minimized to achieve image reconstruction. Several iterations successfully implemented ray attenuation correction, resulting in high-quality images. The newly designed algorithm exhibits advantages in reconstruction quality and convergence speed, significantly reducing reconstruction time and memory usage without compromising image quality.

In 2018, Ge et al. [193] constructed a physical model for a Compton γ camera based on a 3D position-sensitive cadmium zinc telluride detector using Geant4 software. Image reconstruction work was performed based on data obtained from Monte Carlo simulations. The researchers applied inversion reconstruction and the maximum likelihood expectation maximization (MLEM) algorithm for image reconstruction.

Additionally, in 2019, Wang [194] simulated a dual-layer Compton imaging system composed of position-sensitive CZT crystals. Using a combined theoretical and simulation-based approach, they conducted theoretical research on several factors that cause scattering angle errors in this imaging system, affecting its angular resolution capabilities. The team calculated the range of scattering angle errors caused by various factors affecting angular resolution for photons of different energies, subsequently proposing an optimized design for the imaging system structure. They provided a rational estimation method for the scattering angle errors in Compton imaging systems.

Furthermore, in 2019, Song et al. [195] investigated a novel non-mechanical collimated Compton camera technology for cancer therapy devices. Their research confirmed that the spatial resolution of the detector is the primary factor influencing the quality of Compton camera imaging. They emphasized the need to select appropriate crystal unit sizes for the detector while balancing detection efficiency.

In 2021, Zhang [196] developed a three-layer Compton camera with two distinct operational modes based on maximum likelihood estimation (MLEM). These two modes leverage single and double Compton scatterings, and the systems with Depth-of-Interaction (DOI) information feedback are introduced. Thereby enhancing the system’s detection efficiency while enabling the simultaneous detection of γ sources of different energies. The system underwent simulation using Geant4 and Python3.9 software, culminating in the detection outcomes of single-point and multi-point sources for both single and double Compton scatterings, as illustrated in Figure 10.

In Figure 10A(a,b), a primary Compton scattering working mode is depicted, with Figure 10A(a) having a DOI, while Figure 10A(b) does not. Figure 10A(c,d) illustrate a secondary Compton scattering working mode, where Figure 10A(c) has a DOI, and Figure 10A(d) does not. It is evident that systems with a DOI, regardless of whether it is a primary or secondary scattering working mode, exhibit superior imaging quality compared to systems without a DOI. From Figure 10B(a,c) and (b,d), we can see that both the one-scattering working mode and the two-scattering working modes can be positioned. The two-scattering operation mode involves more processes and brings more uncertainty from the measurement, so the angular resolution ability is less than the one-scattering operation mode. The information provided suggests that by obtaining the position and deposited energy information of events in the detector, the simulated imaging system achieved an optimal angular resolution of approximately 6.5°. This system enables the simultaneous imaging of sources with different energies through a single measurement. Compared to imaging methods relying solely on the principles of single scattering, the inherent detection efficiency of the system improved by 31.2% under similar conditions.

In 2022, Kim et al. [197] devised an array structure composed of individual virtual Frisch-grid CZT (cadmium zinc telluride) detectors to achieve higher detection efficiency with a relatively low cost and a larger effective volume. They developed a virtual Frisch-grid CZT detector based on this array structure. The output data were processed using the Weighted List-Mode MLEM method, successfully obtaining energy spectra and Compton images. Simulation results indicated that the position of the radioactive source was well determined at various offset angles. However, performance showed a decline with increasing offset angles due to variations in spatial resolution in the *x*, *y* direction and depth.

In summary, both domestic and international research in Compton detection and imaging techniques for CZT detectors primarily focus on algorithm optimization. The aim is to enhance the accuracy of spatial reconstruction of radioactive sources through the combination of iterative algorithm improvements and hardware enhancements.

## 5. Conclusions

CdZnTe, as an outstanding semiconductor detector material, has been continuously esteemed and studied since its discovery. It has been fashioned into various detector configurations. Detectors with array-distributed crystal surfaces have significant importance and utility in the detection and discrimination of γ-rays and X-rays. These detectors, particularly CZT arrays, hold immense promise in nuclear detection applications. Research on CZT array detectors enables the rapid acquisition of high-resolution information about detected objects. 

In recent years, research on CZT array detectors has primarily focused on the following areas:CZT Crystal Research: enhancing crystal performance by improving semiconductor crystal fabrication methods, doping with trace elements, refining etching processes, or applying surface coatings.Optimization of Array Detector Electronics: improving detector readout circuit design, enhancing the performance of electronic components, and increasing detector response speed and energy resolution.Design of Novel CZT Array Detectors: developing and optimizing applications for CZT array detectors, especially in the detection of specific nuclear materials.Enhancement of Array Detector Imaging Algorithms: improving imaging algorithms to achieve better reconstruction results and generate improved 3D images.

CZT array detectors, as semiconductor detectors with significant potential applications, offer numerous areas for further research and optimization, including the refinement of semiconductor crystal performance by enhancing crystal quality through methods such as doping with trace elements like zinc or refining fabrication processes. Additionally, improvements in detector electronics and algorithms are crucial. This involves enhancing the design of readout circuits for charge-sensitive detectors and employing algorithms such as iterative methods or maximum likelihood methods to elevate the quality of generated images. Furthermore, advancements in detector design, such as the utilization of novel array designs and detection modes, hold promise for enhancing overall performance. In conclusion, the CZT array detector exhibits substantial untapped potential, providing ample opportunities for continued exploration and application in future research endeavors.

## Figures and Tables

**Figure 1 sensors-24-00725-f001:**
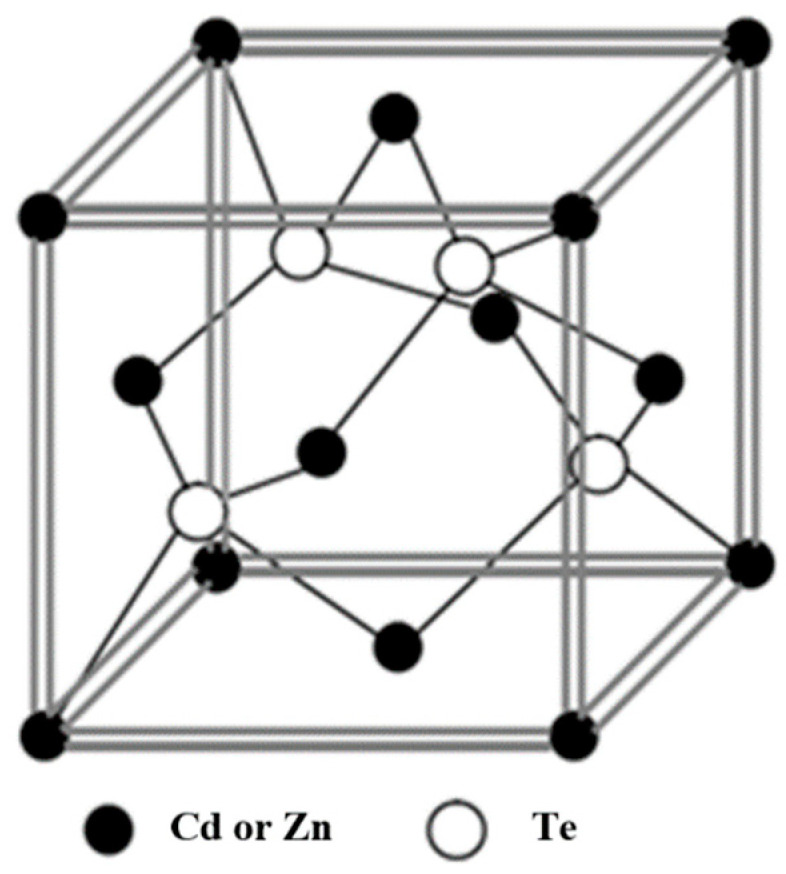
Lattice structure of CZT crystals.

**Figure 2 sensors-24-00725-f002:**
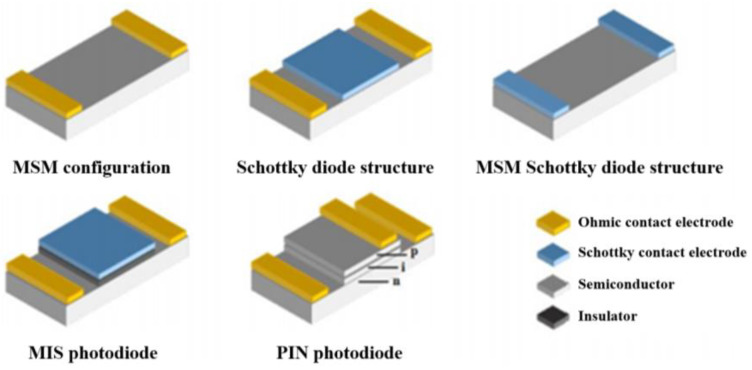
Schematic diagram of semiconductor photodetectors with different configurations [88]. This figure is reprinted with the permission of Springer Nature.

**Figure 3 sensors-24-00725-f003:**
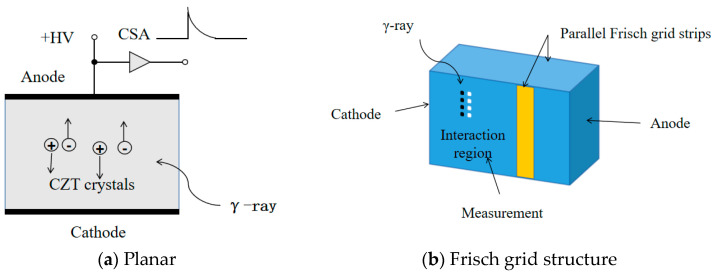
Schematic diagram of CZT detector surface configuration with different unipolar electrode structures [1,93]. (**c**,**e**,**f**) reprinted with the permission of Elsevier.

**Figure 4 sensors-24-00725-f004:**
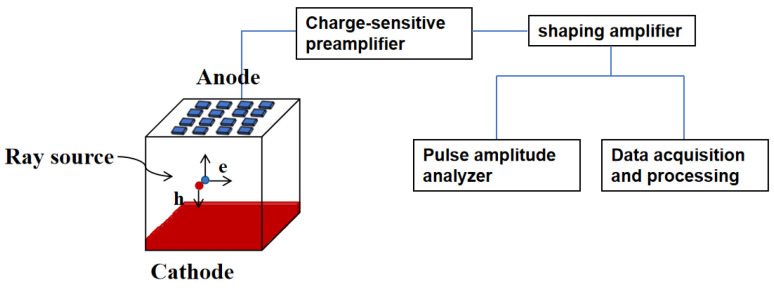
Schematic diagram of the working principle of CZT array detector.

**Figure 5 sensors-24-00725-f005:**
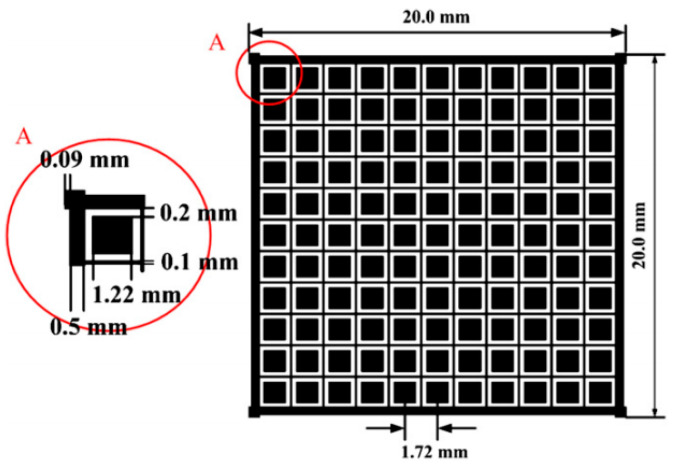
Anode design of pixelated CZT detector for non-collecting steering mesh model [117]. This figure is reprinted with the permission of Elsevier.

**Figure 6 sensors-24-00725-f006:**
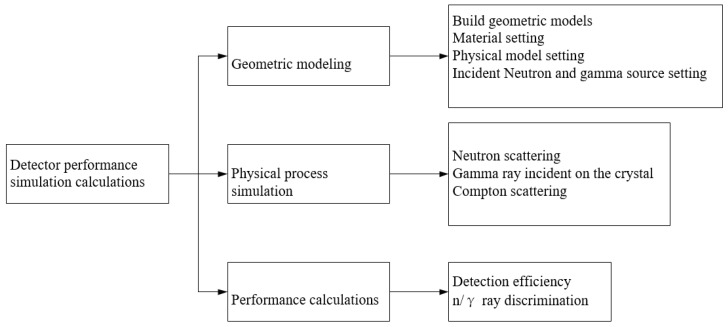
Geant4 simulation calculation flowchart.

**Figure 7 sensors-24-00725-f007:**
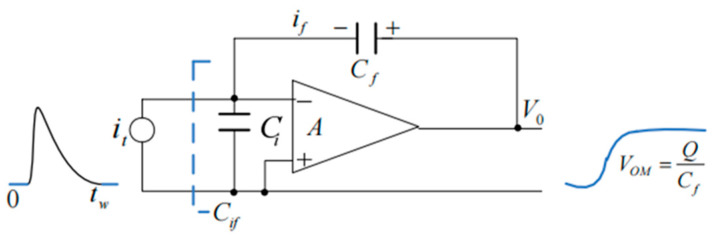
Schematic diagram of the charge-sensitive preamplifier circuit.

**Figure 8 sensors-24-00725-f008:**
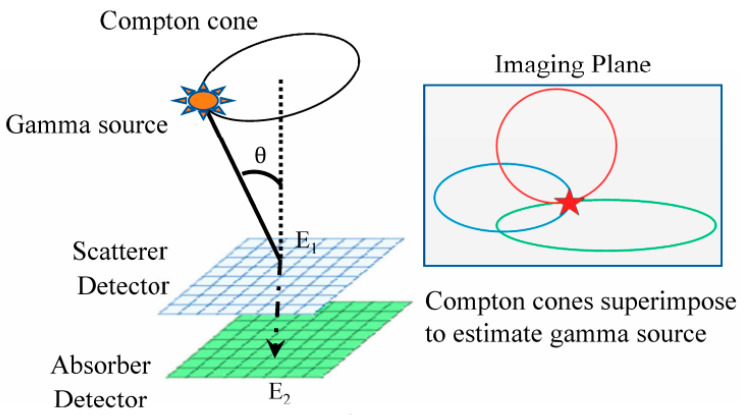
Schematic illustration of a general Compton camera (**left**); Compton cones of each event are superimposed to locate the γ-ray source (**right**) [176].

**Figure 9 sensors-24-00725-f009:**
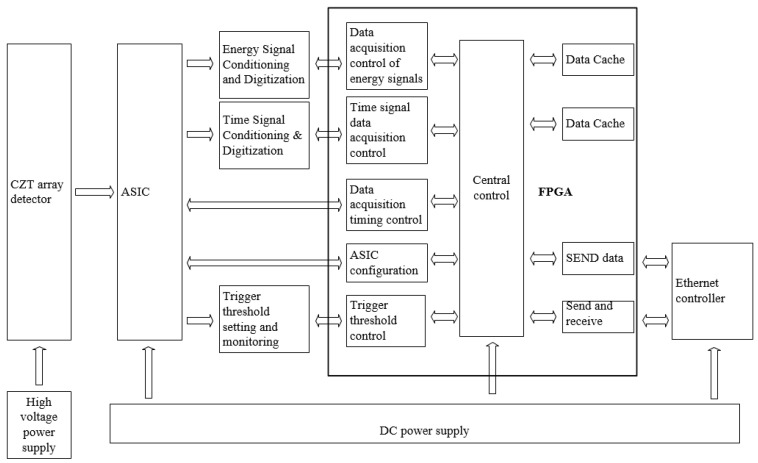
Block diagram of the data acquisition board principle design.

**Figure 10 sensors-24-00725-f010:**
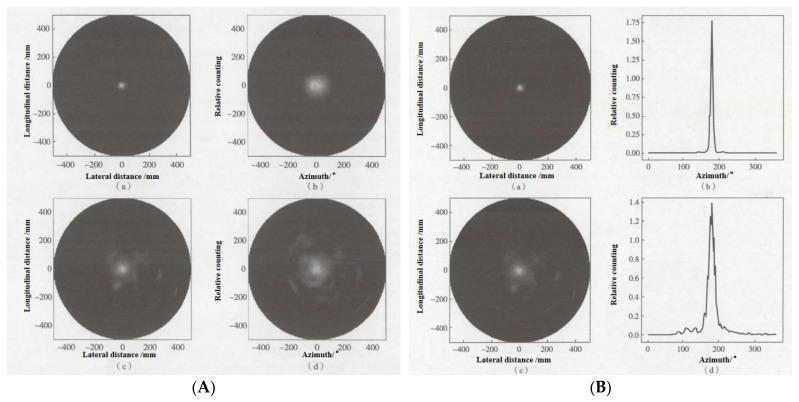
(**A**) Single-point source imaging effect, (**B**) multi-point source imaging effect. Figure 10A(a,b), a primary Compton scattering working mode is depicted, with Figure 10A(a) having a DOI, while Figure 10A(b) does not. Figure 10A(c,d) illustrate a secondary Compton scattering working mode, where Figure 10A(c) has a DOI, and Figure 10A(d) does not. Figure 10B(a,b): secondary Compton scattering working mode. Figure 10B(c,d): secondary Compton scattering working mode.

**Table 1 sensors-24-00725-t001:** Comparison of the properties of different detector materials [12,13,14].

Material	Atomic Number	Density (g/cm^3^)	Bandgap Width (eV)	Ionization Energy (eV)	Resistivity (Ω)	μ_e_τ_e_	μ_h_τ_h_
Si	14	2.33	1.12	3.62	10^4^	>1	1
Ge	32	5.33	0.67	2.96	50	>1	>1
InP	15/49	4.78	1.35	4.2	10^6^	5 × 10^−6^	<2 × 10^−5^
GaAs	33/31	5.32	1.43	4.2	10^7^	10^−5^	10^−6^
HgI_2_	80/53	6.40	2.13	4.2	10^13^	10^−4^	10^−5^
PbI_2_	82/53	6.20	2.3~2.6	4.9	10^12^	10^−6^	10^−7^
TlBr	81/35	7.56	2.68	6.5	10^12^	10^−5^	10^−6^
CdTe	48/52	6.20	1.44	4.43	10^9^	10^−3^	10^−4^
Cd_0.9_Zn_0.1_Te	48/30/52	5.78	1.57	4.64	10^10^~10^11^	10^−3^~10^−2^	10^−5^
Cd_0.8_Zn_0.2_Te	48/30/52	6.02	1.5~2.2	5.0	10^10^~10^11^	10^−3^	10^−6^~10^−5^

**Table 2 sensors-24-00725-t002:** Advantages and disadvantages of different CZT crystal growth methods.

Preparation Method	Advantages	Disadvantages
BV	Simple structure and ease of operation for producing uniform CZT crystals; various Bridgman method variations available as per requirement.	Potential issues of crystal cracking and occurrences of polycrystalline and twinning phenomena
HPB	Provides a high crystal growth rate by using high-pressure inert gas to prevent element evaporation	May still encounter crystal cracking and potential polycrystalline and twinning issues
MVB	Enhanced CZT crystal quality using improved techniques; production of relatively large CZT crystals	Possibility of twinning and grain boundary problems
HB	Achieves a uniform matrix during growth, improving yield and cost-effectiveness	Might result in lower volume resistivity, potentially affecting energy resolution; not suitable for detecting lower energy radiation
THM	Enables continuous crystal growth, resulting in CZT crystals with higher uniformity and lower defect density	Slower crystal growth rates, significant temperature gradients during preparation, which can lead to temperature fluctuations and uneven solute distribution
PVT-VTE	The prepared crystal has high resistivity, small electron drift time, few defects, and good carrier transport performance	The preparation process needs to be further refined to explore its potential

**Table 3 sensors-24-00725-t003:** Relationship between CZT bandgap and Zn content and temperature [71,72,73,74,75,76,77,78].

Empirical Formula	Research Group	Temperature
E*g* (eV) = 1.604 + 0.42*x* + 0.33*x*^2^	Taguchi	9 K
E*g* (eV) = 1.5964 + 0.445*x* + 0.33*x*^2^	Doty	12 K
E*g* (eV) = 1.586 + 0.5006*x* + 0.29692*x*^2^		77 K
E*g* (eV) = 1.4637 + 0.496*x* + 0.2289*x*^2^		300 K
E*g* (eV) = 1.598 + 0.614*x* − 0.116*x*^2^	Polichar	
E*g* (eV) = 1.5 + 0.5*x* + 0.2*x*^2^	Toney	
E*g* (eV) = 1.5045 + 0.631*x* + 0.128*x*^2^	Tobin	
E*g* (eV) = 1.606 + 0.332*x* + 0.462*x*^2^	Hoschl	
E*g* (eV) = (1.494 ± 0.005) + (0.606 ± 0.010)*x* + (0.139 ± 0.010)*x*^2^	Li	

## Data Availability

The raw data supporting the conclusions of this article will be made available by the authors on request.

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
