# Peer review of "Research on the Technological Progress of CZT Array Detectors"

_sensors, 2024, doi:10.3390/s24030725_

Round 1
Reviewer 1 Report
Comments and Suggestions for Authors
The review is wide and very detailed, probably even too wide. The manuscript quality is high.
Small remark concerns eqs. 14 and 15 on page 10. Difference between the equations should be clarified: 93.9% of neutrons captures are accompanied by gamma emission of 0.46 MeV energy and this has to be mentioned. Moreover, these gammas can also give signals in addition to those produces by alphas and recoil nuclei. Therefore, neutron detection by these detectors has to be confirmed.
Reviewer 2 Report
Comments and Suggestions for Authors
Li et al. reviewed the progress of CZT Array Detectors. This review is well written and of interest to the readers. It is recommended to be accepted after minor revision.
1. Some number in the tables and the main text should be superscript or subscript, especially Table 1, please check.
2. In Figure 10, some non-English words exist. Better translate them to English.
3. Some recently published related works about fabrication and performance improvement of detectors should be cited: Nano-Micro Lett. 2023, 15, 128; Materials Futures, 2023, 2(1): 012104; Nano Research, 2023, 16(7), 10256; J. Semicond, 2022, 43(6): 062804.
4. In Table 1, the unit for Resistivity should be provided.
5. Better add some words in Figure 8 to help the readers to understand the content.
Comments on the Quality of English LanguageQuality of English language is high
Reviewer 3 Report
Comments and Suggestions for Authors
In this manuscript entitled “Research on the Technological Progress of CZT Array Detectors”, Li and coauthors have provided a systematic overview on the CdZnTe (CZT) crystals and CZT array radiation detectors. Overall, CZT is a promising material with competitive physical properties, and nuclear detection is important to warrant healthy life and working environment. As such, the topic is worthy of publication. However, in the current stage, I find that there is still room for improvement. Therefore, a revision is recommended. The following comments should be fully taken into account.
1. Page 3, Table 1, the unit of Resistivity is missing. In addition, there are numerous mistakes in terms of the superscript/subscript.
2. Some latest advancement (e.g., Vacuum 2021, 183, 109855; Crystals 2022, 12, 187; Sensors 2023, 23, 2167) in this area has been missed. These studies should be included to provide a more comprehensive panoramagram of the exciting area to the readers.
3. The authors should provide the provenance of the images in the corresponding figure captions, provided the images are taken from the previous studies.
4. All abbreviations (e.g., SHAPER) should be clearly defined at their first appearance. Otherwise, they will cause confusion.
5. In the sentence of “With the advancement of nuclear science and technology, nuclear detection technology is now widely used in people’s daily lives and scientific research”, it would be better to provide some supporting references (Prog. Nucl. Energy 2021, 140, 103918; Adv. Mater. 2023, 10.1002/adma.202304523; Mater. Futures 2023, 2, 012104) in the related area to broaden the readers’ horizons.
6. In the conclusion section, the discussion of future development direction is much too simple. Try to make more detailed discussion to better guide readers to carry out further investigation in the upcoming future.
Reviewer 4 Report
Comments and Suggestions for Authors
Dear Editor,
The authors have investigated CZT array detectors and their performances. The novelty of the manuscript has not been highlighted so I recommend the major revision based on the following comments.
1. It is essential that the innovation of the manuscript is highlighted in the text.
2. The results obtained from this work should be compared with other works.
3. In the introduction section, the text needs to be improved by adding more related works, so the following articles should be reviewed to give a better view to the readers.
Vacuum 183 (2021): 109855, Frontiers of Optoelectronics 5 (2012): 317-321, Hard X-ray, gamma-ray, and neutron detector physics XX. Vol. 10762. SPIE, 2018, Optical and Quantum Electronics 54.3 (2022): 171, Journal of Radiation Protection and Research 45.1 (2020): 35-44, IEEE Transactions on Device and Materials Reliability 21.3 (2021): 389-393.
4. Dark current, sensitivity, linear coefficient, and detectivity have not been calculated.
5. What is the acceptable range for the optical power and wavelength of the detector?
6. What is the advantage of a CZT detector over MCT?
7. What is the resistance of the contacts? Calculate the time response of the detector.
Kind regards,
Round 2
Reviewer 3 Report
Comments and Suggestions for Authors
The authors have addressed all my previous concerns. Now the manuscript can be recommended for publication.
Reviewer 4 Report
Comments and Suggestions for Authors
Dear Editor,
I studied the revised version of the manuscript. According to the response, the manuscript can be accepted for publication in its present form.
Kind regards,